# Inhibited KdpFABC transitions into an E1 off-cycle state

**Jakob M Silberberg**[1†], **Charlott Stock**[1†‡], **Lisa Hielkema**[2†], **Robin A Corey**[3], **Jan Rheinberger**[2], **Dorith Wunnicke**[1], **Victor RA Dubach**[2], **Phillip J Stansfeld**[4], **Inga Hänelt**[1*], **Cristina Paulino**[2*]

[1]Institute of Biochemistry, Biocenter, Goethe University Frankfurt, Frankfurt, Germany; [2]Department of Structural Biology, Groningen Biomolecular Sciences and Biotechnology Institute, University of Groningen, Groningen, Netherlands; [3]Department of Biochemistry, University of Oxford, Oxford, United Kingdom; [4]School of Life Sciences & Department of Chemistry, University of Warwick, Coventry, United Kingdom

*For correspondence:
haenelt@biochem.uni-frankfurt.de (IH);
c.paulino@rug.nl (CP)

[†]These authors contributed equally to this work

**Present address:** [‡]DANDRITE, Nordic EMBL Partnership for Molecular Medicine, Department of Molecular Biology and Genetics, Aarhus University, Aarhus, Denmark

**Competing interest:** The authors declare that no competing interests exist.

**Abstract** KdpFABC is a high-affinity prokaryotic K[+] uptake system that forms a functional chimera between a channel-like subunit (KdpA) and a P-type ATPase (KdpB). At high K[+] levels, KdpFABC needs to be inhibited to prevent excessive K[+] accumulation to the point of toxicity. This is achieved by a phosphorylation of the serine residue in the $TGES_{162}$ motif in the A domain of the pump subunit KdpB ($KdpB_{S162\text{-}P}$). Here, we explore the structural basis of inhibition by $KdpB_{S162}$ phosphorylation by determining the conformational landscape of KdpFABC under inhibiting and non-inhibiting conditions. Under turnover conditions, we identified a new inhibited KdpFABC state that we termed E1P tight, which is not part of the canonical Post-Albers transport cycle of P-type ATPases. It likely represents the biochemically described stalled E1P state adopted by KdpFABC upon $KdpB_{S162}$ phosphorylation. The E1P tight state exhibits a compact fold of the three cytoplasmic domains and is likely adopted when the transition from high-energy E1P states to E2P states is unsuccessful. This study represents a structural characterization of a biologically relevant off-cycle state in the P-type ATPase family and supports the emerging discussion of P-type ATPase regulation by such states.

## Editor's evaluation

KdpFABC is a bacterial potassium uptake transporter made up of a channel-like subunit (KdpA) and a P-type ATPase (KdpB). When potassium levels are low (< 2 mM), the transporter actively and selectively uptakes potassium, but must be switched off again to prevent excessive potassium accumulation. Here, by using cryo-EM, pulsed EPR measurements and MD simulations the molecular basis KdpFABC for inhibition by phosphorylation has been defined to an arrested (off-state) that is in a distinct conformation from previously determined P-type ATPase structures.

## Introduction

A steady intracellular K[+] concentration is vital for bacterial cells. Various export and uptake systems jointly regulate bacterial K[+] homeostasis when facing rapid changes in the environment (*Diskowski et al., 2015*; *Stautz et al., 2021*). KdpFABC is a primary active K[+] uptake system, which is produced when the extracellular K[+] supply becomes too limited for uptake by less affine translocation systems like KtrAB, TrkAH, or Kup. Due to its high affinity for K[+] ($K_m$ = 2 μM) and its active transport, KdpFABC can pump K[+] into the cell even at steep outward-directed gradients of up to 10[4], thereby guaranteeing

a cytosolic K$^+$ level of about 200 mM and cell survival (*Altendorf et al., 1998*; *Epstein et al., 1993*; *Rhoads and Epstein, 1977*; *Weiden et al., 1967*).

The heterotetrameric KdpFABC complex comprises four subunits, namely the channel-like KdpA, a member of the superfamily of K$^+$ transporters (SKT) (*Durell et al., 2000*), the P-type ATPase KdpB (*Hesse et al., 1984*), the lipid-like stabilizer KdpF (*Gassel et al., 1999*), and KdpC, whose function is still unknown. KdpB consists of a transmembrane domain (TMD) and the characteristic cytoplasmic nucleotide-binding (N), phosphorylation (P), and actuator (A) domains. Analogous to all P-type ATPases, KdpB follows a Post-Albers reaction scheme, switching between E1 and E2 states that provide alternating access to the substrate-binding site during turnover (*Albers, 1967*; *Huang et al., 2017a*; *Post et al., 1972*; *Silberberg et al., 2021*; *Stock et al., 2018*; *Sweet et al., 2021*). Whilst in its outward-open E1 state, KdpFABC binds ATP in the N domain and takes up K$^+$ ions via the selectivity filter in KdpA, which progress into the intersubunit tunnel connecting KdpA and KdpB. Binding of ATP causes rearrangements of the cytoplasmic domains that result in nucleotide coordination between the N and P domains in the E1·ATP state. The γ-phosphate of ATP is coordinated in close proximity to the highly conserved KdpB$_{D307}$ of the P domain (all residue numbers refer to *Escherichia coli* KdpFABC). Upon binding of K$^+$ to the canonical substrate-binding site (CBS) of KdpB, KdpB$_{D307}$ cleaves off the ATP γ-phosphate via a nucleophilic attack, leading to the autophosphorylation of KdpFABC and progression to the energetically unfavorable E1P state. ATP cleavage releases the N domain from the P domain, allowing relaxing rearrangements of the cytoplasmic domains that convert KdpFABC to the inward-open E2P state, in which K$^+$ is released from the CBS to the cytoplasm due to conformational changes in KdpB's TMD. Finally, KdpB$_{D307}$ is dephosphorylated by a water molecule coordinated by KdpB$_{E161}$ of the TGES$_{162}$ loop in the A domain, recycling KdpFABC via the non-phosphorylated E2 state back to its E1 ground state (*Huang et al., 2017a*; *Pedersen et al., 2019*; *Stock et al., 2018*; *Sweet et al., 2021*). Thus, the relative orientation of the three cytoplasmic domains to each other and their nucleotide state (nucleotide-free, nucleotide-bound, or phosphorylated) are crucial for the assignment of catalytic states (*Bublitz et al., 2010*; *Dyla et al., 2020*). Notably, while the general catalytic reactions and conformational arrangements of the N, P, and A domains follow the conventional Post-Albers cycle observed for other P-type ATPases, the alternating access of the substrate-binding site in the KdpFABC complex is inverted to accommodate KdpFABC's unique intersubunit transport mechanism involving KdpA and KdpB (*Damnjanovic et al., 2013*; *Silberberg et al., 2021*; *Stock et al., 2018*).

Being a highly efficient emergency K$^+$ uptake system, KdpFABC needs to be tightly regulated in response to changing K$^+$ conditions, as both too low and too high potassium concentrations would be toxic (*Roe et al., 2000*; *Stautz et al., 2021*). At low K$^+$ conditions, transcription of the *kdpFABC* operon is activated by the K$^+$-sensing KdpD/KdpE two-component system (*Polarek et al., 1992*). Further, post-translational stimulation is conferred by cardiolipin, whose concentration increases as a medium-term response to K$^+$ limitation (*Schniederberend et al., 2010*; *Silberberg et al., 2021*). Once K$^+$ stress has abated ([K$^+_{external}$] > 2 mM), the membrane-embedded KdpFABC is rapidly inhibited to prevent excessive uptake of K$^+$ (*Roe et al., 2000*). This is achieved by a post-translational phosphorylation of KdpB$_{S162}$ (yielding KdpFAB$_{S162-P}$C), which is part of the highly conserved TGES$_{162}$ motif of the A domain (*Huang et al., 2017a*; *Sweet et al., 2020*). In the crystal structure of KdpFABC [5MRW], phosphorylated KdpB$_{S162}$ forms salt bridges with KdpB$_{K357}$ and KdpB$_{R363}$ of the N domain and adopts an unusual E1 conformation. It was suggested that the salt bridge formation inhibits KdpFABC by locking the complex in this state (*Huang et al., 2017a*). However, the salt bridges were shown to be non-essential to the inhibition mechanism, leaving the role of this conformation unclear (*Stock et al., 2018*; *Sweet et al., 2020*). Recent functional studies showed that KdpB$_{S162}$ phosphorylation stalls the complex in an intermediate E1P state, preventing the transition to the inward-open E2P state (*Sweet et al., 2020*).

Here, we set out to address the structural basis for KdpFABC inhibition by KdpB$_{S162}$ phosphorylation. The conformational landscape of KdpFABC was probed under different conditions by cryo-EM, yielding 10 structures representative of 6 distinct states that describe the conformational spectrum of the KdpFABC catalytic cycle and resolve the effect of the inhibitory phosphorylation on the conformational plasticity of the complex. Distinct states were further characterized by pulsed electron paramagnetic resonance (EPR) measurements and molecular dynamics (MD) simulations to decipher how KdpB$_{S162}$ phosphorylation leads to the inhibition of the complex in the high-energy E1P intermediate.

**Table 1.** Conformational landscape of KdpFABC resolved by cryo-EM.
Four samples were prepared for cryo-EM analysis. Structural models were built for all maps with a resolution of 4 Å or better.

| Protein sample | Condition | State/resolution | | | | | | | |
| | | E1 apo open | | | | | | | |
| | | Substate 1 | Substate 2 | E1 apo tight | E1·ATP$_{early}$ | E1P·ADP | E1P tight | E2P |
| KdpFAB$_{S162-P}$C | 50 mM KCl 2 mM ATP ('turnover') | | | | 3.5 Å | 3.1 Å | 3.4 Å | |
| | 1 mM KCl 2 mM VO$_4^{3-}$ | | | | | | 3.3 Å | 7.4 Å |
| KdpFAB$_{S162-P/D307N}$C | 50 mM KCl | 3.5 Å | 3.7 Å | 3.4 Å | | | | |
| KdpFAB$_{S162A}$C | 50 mM KCl 2 mM ATP ('turnover') | | | | 3.7 Å | | | 4.0 Å |

The online version of this article includes the following source data for table 1:

**Source data 1.** Cryo-EM data collection, refinement, and validation statistics.

## Results

Previous structural studies of KdpFABC were all conducted in the presence of inhibitors with the aim of stabilizing and obtaining distinct states of the catalytic cycle (*Huang et al., 2017a*; *Silberberg et al., 2021*; *Stock et al., 2018*; *Sweet et al., 2021*). To describe the structural effects of KdpFABC inhibition by KdpB$_{S162}$ phosphorylation, we prepared cryo-EM samples of KdpFABC variants under conditions aimed at covering the conformational landscape of both the non-phosphorylated, active and the phosphorylated, inhibited complex (*Table 1*; *Table 1—source data 1*). For this, all KdpFABC variants were produced in *Escherichia coli* at high K$^+$ concentration, which is known to lead to the inhibitory KdpB$_{S162}$ phosphorylation (*Sweet et al., 2020*). KdpFAB$_{S162A}$C, a non-phosphorylatable KdpB$_{S162}$ variant, and wild-type KdpFABC (KdpFAB$_{S162-P}$C) were analyzed under turnover conditions, that is in the presence of saturating KCl and Mg$^{2+}$-ATP concentrations, to gain insights into the dynamic conformational landscape adopted by the complex during catalysis. To supplement these samples, inhibited WT KdpFAB$_{S162-P}$C was prepared in the presence of orthovanadate, known to normally arrest P-type ATPases in an E2P or E2·P$_i$ state. Further, the catalytically inactive variant KdpFAB$_{S162-P/D307N}$C was prepared under nucleotide-free conditions to investigate E1 apo states. In sum, the 10 maps obtained from the four samples cover the entire KdpFABC conformational cycle, except for the highly transient E1P state after ADP release and the E2 state after dephosphorylation (*Table 1—source data 1*). All structures exhibit the same intersubunit tunnel described previously, which varies in length depending on the state (*Figure 1—figure supplement 1*; *Huang et al., 2017a*; *Silberberg et al., 2021*; *Stock et al., 2018*; *Sweet et al., 2021*). Each tunnel is filled with non-protein densities assigned as potassium ions, which vary slightly in number and position between structures (*Figure 1—figure supplement 1B*).

### Non-inhibited KdpFABC transitions through the Post-Albers cycle under turnover conditions

Previous functional and structural studies have shown that non-phosphorylatable KdpFAB$_{S162A}$C can adopt major states of the Post-Albers cycle (*Sweet et al., 2021*; *Sweet et al., 2020*). However, the different states were captured with the help of various state-specific inhibitors, limiting our understanding of KdpFABC's full conformational landscape. To determine the predominant states under turnover conditions, non-phosphorylatable KdpFAB$_{S162A}$C was incubated with 2 mM Mg$^{2+}$-ATP and 50 mM KCl for 5 min at 24°C immediately before plunge freezing and analyzed by cryo-EM. From this dataset, we obtained 'only' two structures of KdpFABC (*Figure 1*; *Table 1*; *Figure 1—figure supplements 2 and 3*; *Table 1—source data 1*).

The first structure was resolved globally at 3.7 Å and derived from ~35% of the initial particle set. The overall orientation of the cytosolic domains is reminiscent of the previously published E1·ATP structures [7NNL], [7NNP], and [7LC3] (root mean square deviations [RMSDs] of cytosolic domains 2.41, 2.61, and 2.69 Å, respectively) (*Figure 1A*; *Figure 1—figure supplement 3E*; *Table 2*; *Silberberg*

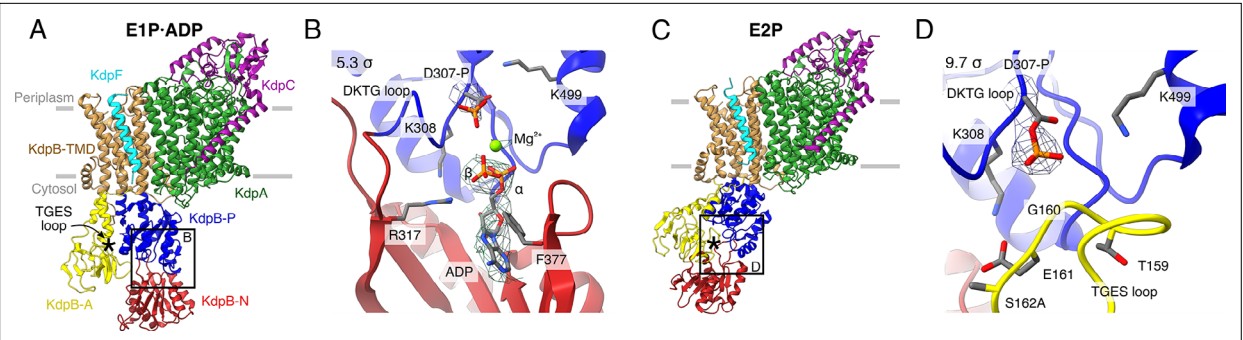

**Figure 1.** Structures of KdpFAB$_{S162A}$C obtained at turnover conditions. Color code throughout the manuscript, unless stated otherwise, is as follows: KdpF in cyan, KdpA in green, KdpC in purple, KdpB transmembrane domain (TMD) in sand with P domain in blue, N domain in red and A domain in yellow. The position of the TGES$_{162}$ loop is denoted by an asterisk. (**A**) E1P·ADP structure, with its nucleotide-binding site (**B**), showing the bound ADP with Mg$^{2+}$. (**C**) E2P structure, with its nucleotide-binding site (**D**), showing the catalytically phosphorylated KdpB$_{D307}$ (P domain), and the TGES$_{162}$ loop (A domain) in close proximity. Densities are shown at the indicated $\sigma$ level.

The online version of this article includes the following figure supplement(s) for figure 1:

**Figure supplement 1.** Comparison of transmembrane domains (TMDs) of all KdpFABC structures obtained in this study.

**Figure supplement 2.** Cryo-EM analysis of the KdpFAB$_{S162A}$C complex under turnover conditions, resulting in E1P·ADP and E2P states.

**Figure supplement 3.** Cryo-EM validation of KdpFAB$_{S162A}$C under turnover conditions.

**Figure supplement 4.** A and P domain movements during the E1P/E2P transition.

---

*et al., 2021*; *Sweet et al., 2021*). However, closer inspection of the N domain reveals no density around the expected γ-phosphate of ATP, whereas additional density is observed near KdpB$_{D307}$. We interpret that ADP, instead of ATP, is bound and the catalytic aspartate KdpB$_{D307}$ has been phosphorylated (*Figure 1B*). An additional density observed near the phosphorylated KdpB$_{D307}$ coincides with a Mg$^{2+}$ ion in the E1P·ADP SERCA structure [1T5T] (*Sørensen et al., 2004*), and has thus likewise been assigned as a coordinating Mg$^{2+}$ ion. Altogether, this conformation represents an E1P·ADP state, following phosphorylation from ATP but preceding the release of ADP. The second structure obtained from the KdpFAB$_{S162A}$C sample under turnover conditions was resolved globally at 4.0 Å from ~28% of the initial particle set. In the respective cryo-EM map, a density is observable immediately adjacent to KdpB$_{D307}$ and the TGES$_{162}$ loop is in close proximity, which is the hallmark of an E2P or E2·P$_i$ state (*Figure 1C, D*). The overall orientation of the cytosolic domains more closely resembles that of the E2·P$_i$ structures (RMSD of 2.66 and 2.4 Å for [7BH2] and [7BGY]) than that of the E2P structure (RMSDs of 11 Å for [7LC6]) (*Figure 1—figure supplement 3J*; *Table 2*; *Sweet et al., 2021*). However, on closer inspection, the density fits better to a to a covalently bound phosphate than to a coordinated P$_i$ (*Figure 1D*). In light of the ambiguity at the given resolution, we have assigned this structure to an E2P state which might be transitioning to an E2·P$_i$. In comparison to the E1P·ADP state, the A domain in the E2P state has undergone a tilt of 60° and a rotation of 64° around the P domain, positioning the TGES$_{162}$ loop to dephosphorylate KdpB$_{D307}$ in the P domain, while the P domain is also tilted by 40° (*Figure 1—figure supplement 4*).

The E1P·ADP and E2P structures obtained for KdpFAB$_{S162A}$C confirm that, in the absence of the inhibitory KdpB$_{S162}$ phosphorylation, KdpFABC progresses through the entire Post-Albers cycle under turnover conditions, with the ADP release in the E1 state and the orthophosphate release in the E2 state likely being the rate-limiting steps that lead to an accumulation of the observed states.

## Inhibited E1P KdpFABC adopts an off-cycle state

To study the structural implications of KdpFABC inhibition by KdpB$_{S162}$ phosphorylation, we next analyzed the conformational landscape of WT KdpFAB$_{S162-P}$C under turnover conditions. To our surprise, the dataset disclosed a higher degree of conformational variability than KdpFAB$_{S162A}$C, yielding three distinct cryo-EM structures (*Figure 2*; *Table 1*; *Figure 2—figure supplements 1 and 2*; *Table 1—source data 1*). The obtained structures show significant deviations in the cytoplasmic region of KdpB, indicative of different positions in the Post-Albers cycle.

**Table 2.** Root mean square deviations (RMSDs) of cytosolic domains of all available structures of KdpFABC. For conformational comparison, RMSDs were calculated for the cytosolic domains of KdpB (residues 89–215, 275–569), as they feature the largest and, for this study, most relevant conformational differences. Structures were superimposed on KdpA, which is rigid across all conformational states. Structural comparisons involving the crystal structure [5MRW] were performed using superpositions on KdpA (column 5MRW A) and on the TM domain of KdpB (residues 1–88, 216–275, 570–682; column 5MRW B) to compensate for structural deviations in the transmembrane domain (TMD) that skew the structural alignment of the KdpB subunit. Shading indicates the RMSD range for each alignment as follows: red, above 15 Å, orange, 10-15 Å, yellow, 6-10 Å, white, 3-6 Å, green, 0-3 Å. Structures obtained in this study are highlighted in bold.

| State | PDB | 6HRA | 7BH1 | **7ZRI** | 7ZRJ | 7LC3 | 7NNL | 7NNP | **7ZRG** | **7ZRM** | **7ZRK** | 7LC6 | **7ZRL** | 6HRB | 7BGY | 7BH2 | **7ZRH** | **7ZRE** | **7ZRD** | 5MRW A | 5MRW B |
|---|---|---|---|---|---|---|---|---|---|---|---|---|---|---|---|---|---|---|---|---|---|
| | 6HRA (3.7 Å) | | 9.67 | 8.28 | 4.14 | 10.31 | 9.69 | 10.02 | 8.59 | 9.32 | 9.50 | 17.15 | 25.42 | 24.91 | 25.01 | 25.53 | 14.25 | 13.25 | 13.07 | 14.29 | 12.06 |
| | 7BH1 (3.4 Å) | 9.67 | | 5.65 | 8.16 | 15.66 | 15.06 | 15.39 | 10.40 | 14.43 | 14.79 | 22.25 | 30.81 | 30.36 | 30.40 | 31.00 | 14.78 | 14.84 | 14.67 | 18.00 | 15.26 |
| | **7ZRI** (3.5 Å) | 8.28 | 5.65 | | 6.36 | 14.83 | 14.14 | 14.47 | 9.61 | 13.49 | 13.86 | 22.58 | 31.33 | 30.85 | 30.87 | 31.43 | 15.11 | 15.64 | 15.52 | 18.50 | 15.58 |
| E1 apo open | **7ZRJ** (3.7 Å) | 4.14 | 8.16 | 6.36 | | 12.27 | 11.57 | 11.91 | 8.93 | 11.06 | 11.34 | 18.65 | 27.15 | 26.67 | 26.72 | 27.23 | 14.99 | 14.41 | 14.23 | 16.04 | 13.54 |
| | 7LC3 (3.2 Å) | 10.31 | 15.66 | 14.83 | 12.27 | | 2.10 | 1.96 | 7.50 | 2.69 | 2.44 | 17.84 | 25.08 | 24.65 | 24.63 | 25.37 | 18.67 | 17.54 | 17.29 | 17.30 | 15.01 |
| | 7NNL (3.1 Å) | 9.69 | 15.06 | 14.14 | 11.57 | 2.10 | | 1.75 | 6.86 | 2.41 | 2.17 | 17.95 | 25.36 | 24.92 | 24.91 | 25.62 | 18.28 | 17.25 | 17.00 | 17.19 | 14.85 |
| | 7NNP (3.2 Å) | 10.02 | 15.39 | 14.47 | 11.91 | 1.96 | 1.75 | | 7.23 | 2.61 | 2.34 | 17.84 | 25.16 | 24.72 | 24.70 | 25.42 | 18.42 | 17.34 | 17.11 | 17.14 | 14.81 |
| E1·ATP | **7ZRG** (3.5 Å) | 8.59 | 10.40 | 9.61 | 8.93 | 7.50 | 6.86 | 7.23 | | 6.13 | 6.57 | 20.08 | 28.13 | 27.69 | 27.68 | 28.40 | 16.59 | 16.21 | 16.01 | 17.80 | 15.33 |
| | **7ZRM** (3.7 Å) | 9.32 | 14.43 | 13.49 | 11.06 | 2.69 | 2.41 | 2.61 | 6.13 | | 2.12 | 18.25 | 25.75 | 25.32 | 25.32 | 26.03 | 18.00 | 17.05 | 16.81 | 17.25 | 14.91 |
| E1P·ADP | **7ZRK** (3.1 Å) | 9.50 | 14.79 | 13.86 | 11.34 | 2.44 | 2.17 | 2.34 | 6.57 | 2.12 | | 17.84 | 25.32 | 24.90 | 24.88 | 25.59 | 18.23 | 17.20 | 16.96 | 17.24 | 14.86 |
| E2P | 7LC6 (3.7 Å) | 17.15 | 22.25 | 22.58 | 18.65 | 17.84 | 17.95 | 17.84 | 20.08 | 18.25 | 17.84 | | 11.03 | 10.44 | 10.32 | 10.71 | 24.86 | 22.44 | 22.32 | 19.25 | 19.44 |
| E2P/E2·Pi | **7ZRL** (4.0 Å) | 25.42 | 30.81 | 31.33 | 27.15 | 25.08 | 25.36 | 25.16 | 28.13 | 25.75 | 25.32 | 11.03 | | 2.52 | 2.43 | 2.66 | 30.42 | 27.40 | 27.30 | 22.80 | 23.23 |
| | 6HRB (4.0 Å) | 24.91 | 30.36 | 30.85 | 26.67 | 24.65 | 24.92 | 24.72 | 27.69 | 25.32 | 24.90 | 10.44 | 2.52 | | 1.98 | 2.18 | 30.05 | 27.05 | 26.96 | 22.42 | 22.93 |
| | 7BGY (2.9 Å) | 25.01 | 30.40 | 30.87 | 26.72 | 24.63 | 24.91 | 24.70 | 27.68 | 25.32 | 24.88 | 10.32 | 2.43 | 1.98 | | 1.74 | 30.18 | 27.20 | 27.10 | 22.61 | 22.37 |
| E2·Pi | 7BH2 (3.0 Å) | 25.53 | 31.00 | 31.43 | 27.23 | 25.37 | 25.62 | 25.42 | 28.40 | 26.03 | 25.59 | 10.71 | 2.66 | 2.18 | 1.74 | | 30.76 | 27.77 | 27.68 | 23.11 | 22.86 |
| E1 apo tight | **7ZRH** (3.4 Å) | 14.25 | 14.78 | 15.11 | 14.99 | 18.67 | 18.28 | 18.42 | 16.59 | 18.00 | 18.23 | 24.86 | 30.42 | 30.05 | 30.18 | 30.76 | | 4.84 | 5.01 | 9.85 | 7.53 |
| E1P tight | **7ZRE** (3.4 Å) | 13.25 | 14.84 | 15.64 | 14.41 | 17.54 | 17.25 | 17.34 | 16.21 | 17.05 | 17.20 | 22.44 | 27.40 | 27.05 | 27.20 | 27.77 | 4.84 | | 1.59 | 6.50 | 4.92 |
| | **7ZRD** (3.7 Å) | 13.07 | 14.67 | 15.52 | 14.23 | 17.29 | 17.00 | 17.11 | 16.01 | 16.81 | 16.96 | 22.32 | 27.30 | 26.96 | 27.10 | 27.68 | 5.01 | 1.59 | | 6.51 | 4.87 |
| Crystal Structure | 5MRW A (2.9 Å) | 14.29 | 18.00 | 18.50 | 16.04 | 17.30 | 17.19 | 17.14 | 17.80 | 17.25 | 17.24 | 19.25 | 22.80 | 22.42 | 22.61 | 23.11 | 9.85 | 6.50 | 6.51 | | |
| | 5MRW B (2.9 Å) | 12.06 | 15.26 | 15.58 | 13.54 | 15.01 | 14.85 | 14.81 | 15.33 | 14.91 | 14.86 | 19.44 | 23.23 | 22.93 | 22.37 | 22.86 | 7.53 | 4.92 | 4.87 | | |

The first KdpFAB$_{S162\text{-}P}$C turnover structure, resolved at an overall resolution of 3.5 Å and derived from ~9% of the initial particle set, roughly resembles the E1·ATP structures [7NNL], [7NNP], and [7LC3] (RMSDs of cytosolic domains 6.86, 7.23, and 7.50 Å, respectively) (*Silberberg et al., 2021*; *Figure 2A*; *Figure 2—figure supplement 2E*; *Table 2*). However, a closer analysis of the cytosolic domains and the bound nucleotide reveals significant differences. While ATP is coordinated in a similar fashion as previously observed (*Silberberg et al., 2021*; *Sweet et al., 2021*), the N domain is slightly displaced relative to the P domain, providing more access to the nucleotide-binding site (*Figure 2A, B*; *Figure 2—figure supplement 2E*). We interpret this structure as an E1·ATP state at an early stage of nucleotide binding and refer to it as E1·ATP$_{early}$. By contrast, inhibitors such as AMPPCP

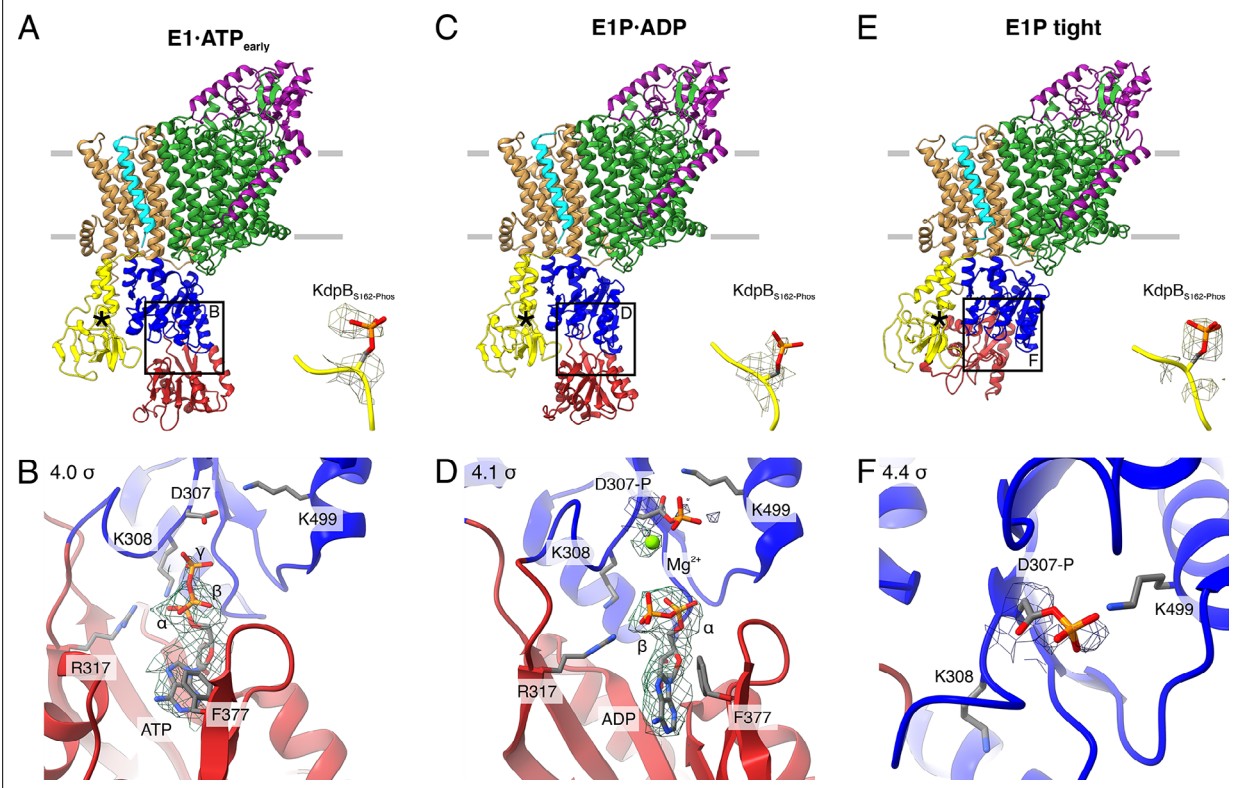

**Figure 2.** Structures of WT KdpFAB$_{S162-P}$C obtained at turnover conditions. (**A**) E1·ATP$_{early}$ structure with the corresponding density of KdpB$_{S162}$ phosphorylation and (**B**) its nucleotide-binding site, showing the bound ATP. (**C**) E1P·ADP structure with the corresponding density of KdpB$_{S162}$ phosphorylation and (**D**) its nucleotide-binding site, showing the bound ADP with Mg$^{2+}$. (**E**) E1P tight structure with the corresponding density for KdpB$_{S162}$ phosphorylation and (**F**) its nucleotide-binding site. Densities are shown at the indicated σ level.

The online version of this article includes the following figure supplement(s) for figure 2:

**Figure supplement 1.** Cryo-EM analysis of the WT KdpFAB$_{S162-P}$C complex under turnover conditions, resulting in the E1P tight, E1P·ADP, and E1·ATP$_{early}$ states.

**Figure supplement 2.** Cryo-EM validation of WT KdpFAB$_{S162-P}$C under turnover.

**Figure supplement 3.** Proximity of A and N domains in E1P·ADP and E1P tight KdpFABC.

**Figure supplement 4.** Cryo-EM analysis of WT KdpFAB$_{S162-P}$C in the presence of orthovanadate, resulting in the E1P tight and E2P state.

**Figure supplement 5.** E1P tight structure of WT KdpFAB$_{S162-P}$C in the presence of orthovanadate.

likely stabilize the latest possible and otherwise transient E1·ATP state immediately before phosphorylation, explaining the discrepancy between the AMPPCP-stabilized [7NNL] and the E1·ATP state obtained here under turnover conditions. Alternatively, this structure could represent an artefactual 'ADP-inhibited' state, in which ADP binds to the N domain and prevents ATP binding. While this is supported by the relatively weak density of the γ-phosphate and the more open conformation, it would be surprising, considering the high excess of ATP used during sample preparation. The second KdpFAB$_{S162-P}$C turnover structure, resolved at an overall resolution of 3.1 Å and representing ~31% of the initial particle set, is very similar to the above-described E1P·ADP state of KdpFAB$_{S162A}$C (RMSD of cytosolic domains 2.12 Å) (*Figure 2C, D*; *Figure 2—figure supplement 2J*; *Table 2*).

While the first two structures represent known states of the catalytic cycle, the third KdpFAB$_{S162-P}$C structure obtained under turnover conditions shows an unusual compact conformation of the cytosolic domains not yet observed in the Post-Albers cycle of other P-type ATPases (*Figure 2E, F*). The structure was resolved at an overall resolution of 3.4 Å and derived from ~14% of the initial particle set. Strikingly, the N domain is closely associated with the A domain, thereby disrupting the nucleotide-binding site between the N and P domains (*Figure 2E, F*; *Figure 2—figure supplement 3*). Closer inspection of the nucleotide-binding site shows that the catalytic aspartate KdpB$_{D307}$ is phosphorylated, but not located in proximity to the TGES$_{162}$ loop of the A domain (*Figure 2F*). This indicates that

the state is adopted after the canonical E1P state, which, in a normal, non-inhibited catalytic cycle, would transition into an E2P state. Due to its compact organization, we termed this state E1P tight.

Previous biochemical studies have shown that KdpB$_{S162}$ phosphorylation inhibits KdpFABC by preventing the transition to an E2 state and stalling it in an E1P state (*Sweet et al., 2020*). In line with these observations, we could not identify any state following E1P in the Post-Albers cycle for KdpFAB$_{S162-P}$C, despite the turnover conditions used. Based on this, we hypothesized that the novel E1P tight state observed represents a non-Post-Albers state that is adopted because KdpFAB$_{S162-P}$C cannot proceed to the E2P state. To further put this to test, we analyzed the conformations adopted by WT KdpFAB$_{S162-P}$C in the presence of the phosphate mimic orthovanadate, which has been shown to trap P-type ATPases in an E2P state and, of all E2 state inhibitors, best mimics the charge distribution of a bound phosphate (*Table 1*; *Figure 2—figure supplements 4 and 5*; *Table 1—source data 1*; *Clausen et al., 2016*; *Pedersen et al., 2019*). Interestingly, KdpFAB$_{S162-P}$C incubated with 2 mM orthovanadate prior to cryo-EM sample preparation did not conform to this behavior. Instead, the major fraction of this sample (~62% of the initial particle set) resulted in an E1P tight state, resolved at 3.3 Å, which is shows no large differences to the E1P tight state obtained under turnover conditions (RMSD of cytosolic domains 1.59 Å) (*Figure 2—figure supplement 4H*; *Table 2*). The position of the orthovanadate coincides with that adopted by the phosphorylated KdpB$_{D307}$, verifying the assignment of this state as a phosphorylated E1P intermediate (*Figure 2—figure supplement 5*). Only a minor fraction of the orthovanadate-stabilized sample (~11% of the initial particle set) adopted an E2P state. Despite the rather poor resolution of 7.4 Å, the catalytic state could be confirmed by comparison of the cryo-EM map with the previously published E2·P$_i$ structure [7BH2] (*Figure 2—figure supplement 4I*; *Sweet et al., 2021*). The minor fraction observed in an E2 state likely represents residual KdpFABC lacking KdpB$_{S162}$ phosphorylation, as it is in good agreement with the residual ATPase activity level found in the sample (*Figure 2—figure supplement 5C*; *Sweet et al., 2020*). The fact that, in the presence of orthovanadate, KdpFAB$_{S162-P}$C adopts the E1P tight state instead of an E2P state strongly

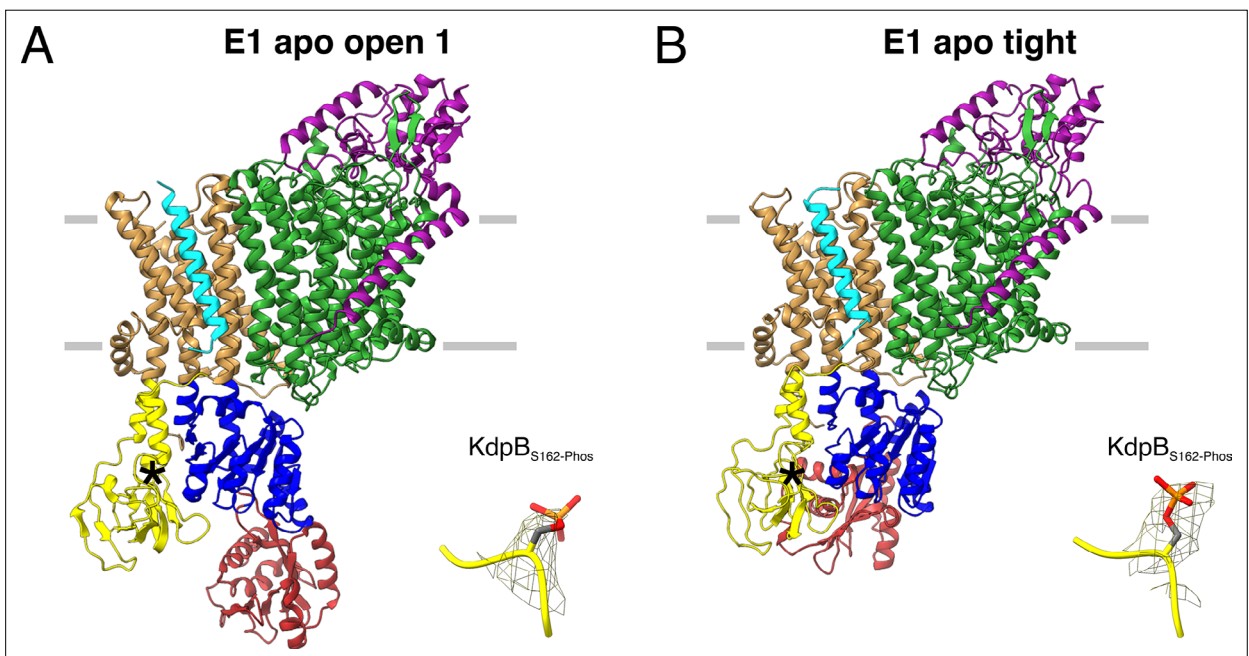

**Figure 3.** E1 apo structures of KdpFAB$_{S162-P/D307N}$C obtained under nucleotide-free conditions. (**A**) E1 apo open substate 1 structure with the corresponding density of KdpB$_{S162}$ phosphorylation. The E1 apo open substate 2 differs by a slightly displaced orientation of the N domain (see also *Figure 3—figure supplements 1 and 2*). (**B**) E1 apo tight structure with the corresponding density of KdpB$_{S162}$ phosphorylation.

The online version of this article includes the following figure supplement(s) for figure 3:

**Figure supplement 1.** Cryo-EM analysis of the KdpFAB$_{S162-P/D307N}$C complex in the absence of nucleotide, resulting in the E1 apo tight and E1 apo open states.

**Figure supplement 2.** Cryo-EM validation of WT KdpFAB$_{S162-P/D307N}$C under nucleotide-free conditions.

supports the idea that the inhibited complex cannot transition into an E2 state but instead adopts an off-cycle E1P state after ADP dissociation.

## A tight state is also formed in the absence of nucleotide

Interestingly, the close association of the N domain with the A domain in the E1P tight conformation also shows some similarities to the E1 crystal structure of KdpFABC [5MRW], the first structure of KdpFABC ever solved (RMSD of cytosolic domains 4.92 Å) (*Table 2*; *Huang et al., 2017a*). Unlike our E1P tight state, the crystal structure is nucleotide-free and does not contain a phosphorylated KdpB$_{D307}$, raising the question how it fits in the conformational landscape of KdpFABC. To further investigate the role of the tight state in the conformational cycle of KdpFABC, we prepared a cryo-EM sample of KdpFAB$_{S162-P/D307N}$C under nucleotide-free conditions. This mutant is catalytically inactive and thus restricted to E1 states preceding phosphorylation in the P domain. In total, we were able to obtain three distinct structures from this preparation, which we assigned to two states relevant to the transport cycle (*Figure 3*, *Table 1*; *Figure 3—figure supplements 1 and 2*; *Table 1—source data 1*).

The first state, composed of 26% of the initial particle set from the nucleotide-free sample, corresponds to the canonical Post-Albers E1 state that precedes ATP binding and resembles the E1 states [6HRA] and [7BH1] reported previously (*Stock et al., 2018*; *Sweet et al., 2021*; *Figure 3A*; *Figure 3—figure supplement 2E*). We have termed this state the E1 apo open state, as the N and P domains are far apart to provide access to the nucleotide-binding site. The cytosolic domains of KdpB reveal a high conformational heterogeneity, evidenced by their lower local resolution (*Figure 3—figure supplement 2B, G*). Focused 3D classification allowed the distinction of two substates, resolved globally at 3.5 and 3.7 Å, differing in the relative position of the N to the P domain (RMSDs between cytosolic domains of 8.28 and 5.65 Å for substate 1 to [6HRA] and [7BH1], respectively, and 4.14 and 8.16 Å for substate 2 to [6HRA] and [7BH1], respectively, with an RMSD of 6.36 Å between substates 1 and 2) (*Figure 3—figure supplement 2J*; *Table 2*). The high degree of flexibility of the N domain in these open states could facilitate nucleotide binding at the start of the Post-Albers cycle. However, it remains unknown whether KdpB can already bind ATP to its N domain in the preceding E2 states, as was shown for other P-type ATPases, which would call the physiological relevance of the E1 apo states into question (*Jensen et al., 2006*).

The second state, featuring one structure resolved globally at 3.4 Å and represented by 19% of the initial particle set, shows a compact conformation of the cytosolic domains with the N and A domains in close contact, providing no space for a nucleotide to bind (*Figure 3B*). This state is similar, but not identical to the one observed in the E1P tight state and [5MRW] (RMSDs of cytosolic domains 4.84 and 7.53 Å, respectively), and we have termed it the E1 apo tight state (*Figure 3—figure supplement 2O*; *Table 2*). The presence of a similar tight conformation in both the E1P and the E1 apo states of KdpB$_{S162-P}$ shows that, while the inhibitory KdpB$_{S162}$ phosphorylation appears to be a prerequisite, the observed close association of the N and A domains can occur before or after the binding and cleavage of ATP.

## The tight state interaction between the N and A domains is itself not the cause of inhibition

Structures with a tight arrangement of the A and N domains were only obtained in cryo-EM samples featuring KdpB$_{S162-P}$. To evaluate the dependency of tight state formation on KdpB$_{S162}$ phosphorylation, we assessed the full conformational flexibility of the N domain using pulsed EPR spectroscopy. Distances between the N and P domains were measured with the labeled residues KdpB$_{A407CR1}$ and KdpB$_{A494CR1}$ (*Figure 4A, B*; *Figure 4—figure supplement 1*). Variants were produced at high K$^+$ concentrations to confer KdpB$_{S162}$ phosphorylation. EPR analysis of KdpFAB$_{S162-P/D307N}$C in the absence of nucleotide resulted in distances that resemble the E1 apo tight (34%) and the E1 apo open (66%) states (*Figure 4B*). This corroborates the cryo-EM results that both tight and open states are adopted in the same sample at nucleotide-free conditions. In contrast, the non-phosphorylatable variant KdpB$_{S162A}$ resulted in a distance distribution showing a significant decrease of the E1 apo tight state (18%). This strongly indicates that the tight state is stabilized by KdpB$_{S162}$ phosphorylation, although it still exists to a small extent even in the absence of the inhibitory phosphorylation.

To inspect what makes the off-cycle tight states energetically preferable, we quantified the cytosolic domain interactions of all E1 states obtained in this study using contact analysis by MD simulation

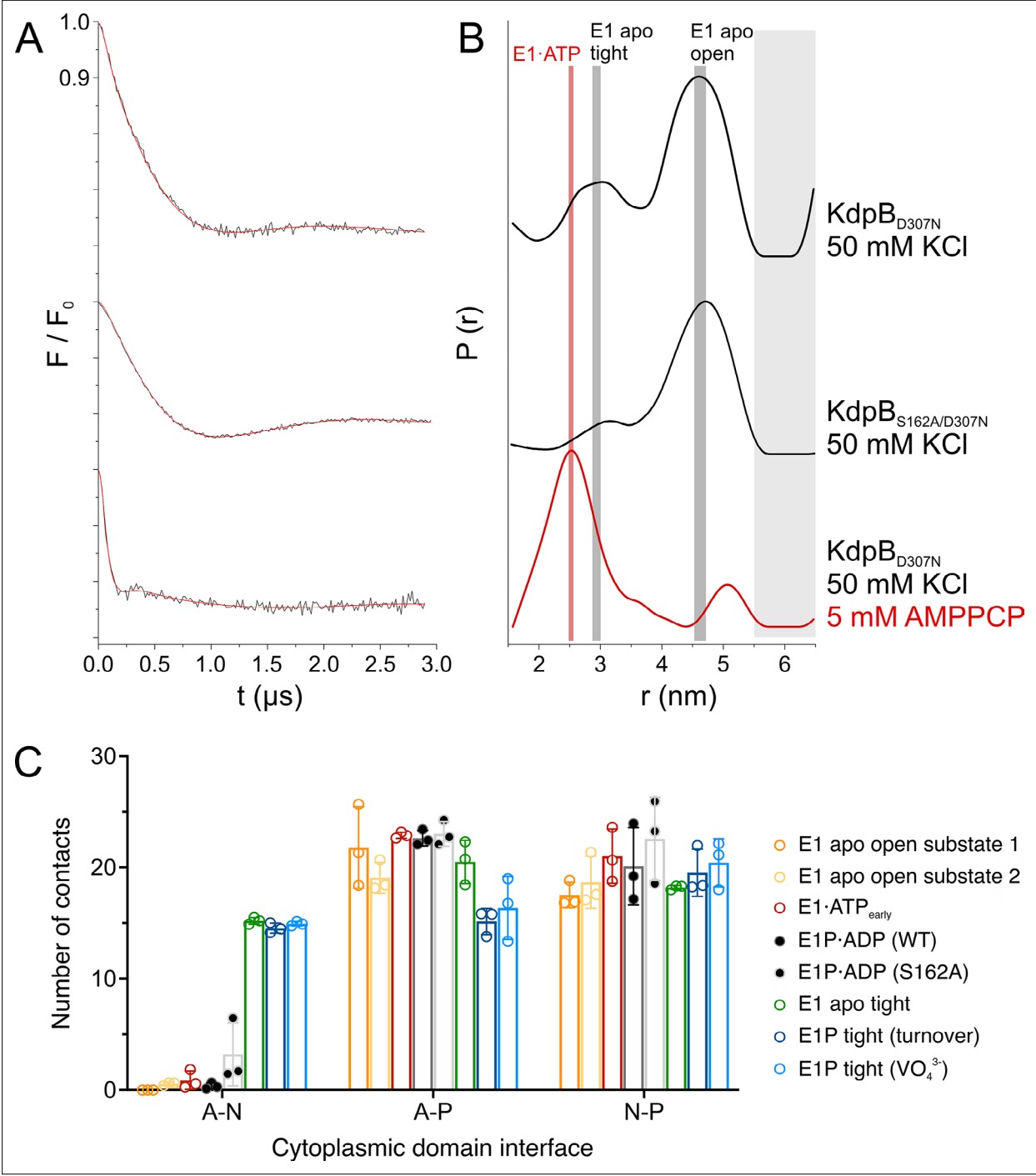

**Figure 4.** KdpB$_{S162-P}$ increases interdomain contacts in E1 tight states. (**A**) Background-corrected dipolar evolution function $F(t)$ with applied fit (red lines) of pulsed EPR measurements. (**B**) Area-normalized interspin distance distribution $P(r)$ obtained by Tikhonov regularization. Two KdpFABC variants were prepared with 50 mM KCl and in the presence (red curve) or absence (black curves) of 5 mM AMPPCP. KdpFAB$_{D307N}$C and KdpFAB$_{S162A/D307N}$C without AMPPCP show two states, with N–P domain distances of 3 and 4.5 nm corresponding to the E1 apo tight and E1 apo open states (dominant state), respectively, as indicated by dark gray background shading. The removal of the inhibitory phosphorylation site in KdpFAB$_{S162A/D307N}$C showed a significant decrease of the distance corresponding to the E1 apo tight state (3 nm) compared to KdpFAB$_{D307N}$C. Addition of AMPPCP to KdpFAB$_{D307N}$C resulted in a single stabilized distance of 2.5 nm, representing the E1·ATP state, as indicated by red background shading. Light gray shaded areas starting at 5.5 nm indicate unreliable distances. (**C**) Contact analysis between the N, P, and A domains for all E1 structures obtained in this study. Average of the number of contacts (>90% contact) between the different domains over 3 × 50 ns molecular dynamics (MD) simulations (see also *Figure 4—figure supplement 2* for details of identified high-contact interactions). Interactions between A and P or N and P domains remain consistent across all states, while interactions between A and N domains are increased only in E1 tight states.

*Figure 4 continued on next page*

*Figure 4 continued*

The online version of this article includes the following figure supplement(s) for figure 4:

**Figure supplement 1.** Pulsed EPR measurements of KdpFABC variants in the absence and presence of AMPPCP.

**Figure supplement 2.** KdpB$_{S162-P}$ mediates stabilization of the E1P tight state.

(*Figure 4C*; *Figure 4—figure supplement 2*). While no large differences are observed for the contacts between A and P or N and P domains, the off-cycle E1 tight states (E1 apo tight, E1P tight) are the only ones to show a higher number of interactions between the A and N domains. In fact, they share a nearly identical number of contacts, indicating that the A and N domains move in a concerted manner, similar to previous observations by MD simulations (*Dubey et al., 2021*). The most evident interaction seen in all three tight states is formed by the salt bridges between the phosphate of KdpB$_{S162-P}$ in the A domain and KdpB$_{K357/R363}$ in the N domain, which were first described in [5MRW] (*Huang et al., 2017a*). However, functional studies have shown that these salt bridges are not essential for the inhibition (*Sweet et al., 2020*), and neither are they sufficient to fully arrest a tight state, as shown here. In agreement with this, EPR measurements with KdpFAB$_{S162-P/D307N}$C in the presence of the ATP analogue AMPPCP result in a distance distribution that shows a single and well-defined distance, which corresponds to the E1·ATP state. Hence, the salt bridges and the enhanced interaction platform between the A and N domain seen in the tight states likely have a stabilizing role, but can be easily broken and are themselves not the main cause of inhibition.

## The E1P tight state is the consequence of an impaired E1P/E2P transition

As the compact arrangement found in the E1 tight states itself is not the determining factor of inhibition, the question remains how K$^+$ transport is blocked in KdpFAB$_{S162-P}$C and what role the E1P tight state plays. In a non-inhibited transport cycle, KdpFABC rapidly relaxes from the high-energy E1P state into the E2P state. To identify what structural determinants enable the arrest before this transition, we compared the E1P tight structure with the other structures obtained in this study (*Figure 5*). The E1P tight state features a tilt of the A domain by 26° in helix 4 (KdpB$_{198–208}$) that is not found in the other E1 states, including the E1 apo tight state (*Figure 5B, C* – arrow 1). This tilt is reminiscent of the movement the A domain undergoes during the E1P/E2P transition (60° tilt, *Figure 1—figure supplement 4*), but it is not as far and does not feature the rotation around the P domain (*Figure 5—figure supplement 1A–C*). Moreover, the P domain does not show the tilt observed in the normal E1P/E2P transition (*Figure 5—figure supplement 1D*). Notably, these are the movements that bring KdpB$_{D307-P}$ in close proximity of KdpB$_{S162}$ in the catalytic cycle (*Figure 5—figure supplement 2*, *Video 1*). In an inhibited state, where both side chains are phosphorylated, such a transition is most likely impaired due to the large charge repulsion. Moreover, a comparison of the E1P tight structure with the E1P·ADP structure, its most immediate precursor in the conformational cycle obtained, reveals a number of significant rearrangements within the P domain (*Figure 5B, C*). First, Helix 6 (KdpB$_{538–545}$) is partially unwound and has moved away from helix 5 toward the A domain, alongside the tilting of helix 4 of the A domain (*Figure 5B, C* – arrow 2). Second, and of particular interest, are the additional local changes that occur in the immediate vicinity of the phosphorylated KdpB$_{D307}$. In the E1P·ADP structure, the catalytic aspartyl phosphate, located in the D$_{307}$KTG signature motif, points toward the negatively charged KdpB$_{D518/D522}$. This strain is likely to become even more unfavorable once ADP dissociates in the E1P state, as the Mg$^{2+}$ associated with the ADP partially shields these clashes. The ensuing repulsion might serve as a driving force for the system to relax into the E2 state in the catalytic cycle. By contrast, the D$_{307}$KTG loop is largely uncoiled in the E1P tight state, with the phosphorylated KdpB$_{D307}$ pointing in the opposite direction, releasing this electrostatic strain (*Figure 5C* – arrow 4). This orientation is further stabilized by a salt bridge formed with KdpB$_{K499}$. The uncoiling in the E1P tight state is likely mediated by the movement of the N domain toward the A domain, as the N domain is directly connected to the D$_{307}$KTG loop (*Figure 5C* – arrow 3). Altogether, we propose that, in the presence of the inhibitory KdpB$_{S162-P}$, the high-energy E1P·ADP state can no longer transition into an E2P state after release of ADP and Mg$^{2+}$. As a consequence, the conformational changes observed in the E1P tight state likely ease the electrostatic tensions of the phosphorylated P domain and stall the system in a 'relaxed' off-cycle state.

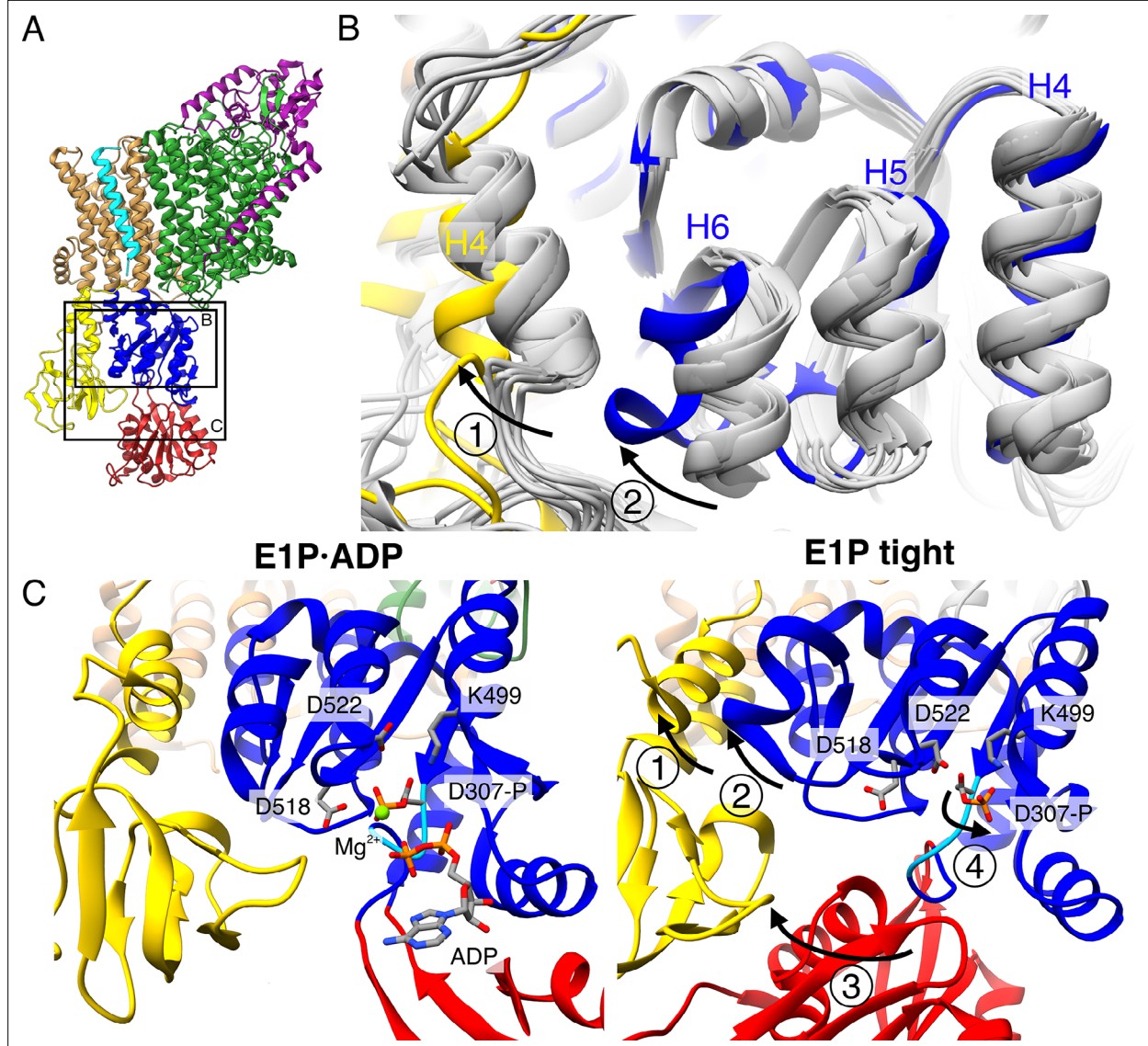

**Figure 5.** Structural rearrangements in the E1P tight state facilitate KdpFAB$_{S162-P}$C stalling. (**A**) Structure of KdpFAB$_{S162-P}$C in the E1P·ADP state obtained at turnover conditions. (**B**) Overlay of all E1 structures determined in this study (in gray) shows helix rearrangements particular to the E1P tight state (in color). Arrow 1 indicates how A domain helix 4 tilts away from the transmembrane domain (TMD) by 26°; Arrow 2 indicates how P domain helix 6 moves along with A domain helix 4, slightly uncoiling and shifting away from the remaining P domain. (**C**) Vicinity of the catalytic aspartyl phosphate (KdpB$_{D307-P}$) in the KdpFAB$_{S162-P}$C E1P·ADP and E1P tight structures, showing rearrangements in the P domain to ease tensions of the E1P state. In the high-energy E1P·ADP state, KdpB$_{D307-P}$ shows strong electrostatic clashes with KdpB$_{D518/D522}$ (2.9 and 3.5 Å, respectively). These are alleviated by rearrangements in the E1P tight state. Arrow 3 indicates the movement of the N domain toward the A domain in the tight state, which pulls on the D$_{307}$KTG loop (light blue) containing the aspartyl phosphate to uncoil and reorient it. Arrow 4 indicates how this rearrangement reorients the phosphorylated KdpB$_{D307-P}$ away from KdpB$_{D518/D522}$ (7.8 and 4.1 Å, respectively) to form a salt bridge with KdpB$_{K499}$ (3.9 Å).

The online version of this article includes the following figure supplement(s) for figure 5:

**Figure supplement 1.** Stalling of A and P domain movements of the E1P/E2P transition in the E1P tight state.

**Figure supplement 2.** Electrostatic repulsion of KdpB$_{S162-P}$ and KdpB$_{D307-P}$ during the E1P/E2P transition.

## Discussion

We set out to deepen our understanding of the structural basis of KdpFABC inhibition via KdpB$_{S162}$ phosphorylation by sampling the conformational landscape under various conditions by cryo-EM. The 10 cryo-EM maps of KdpFABC, which represent six distinct catalytic states, cover close to the full conformational landscape of KdpFABC and, most importantly, uncover a non-Post-Albers regulatory

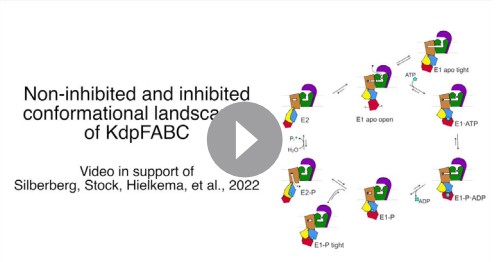

**Video 1.** Illustration of the non-inhibited and inhibited conformational landscape adopted by KdpFABC during its transport cycle, including the transition into the E1P-tight off-cycle state.

https://elifesciences.org/articles/80988/figures#video1

off-cycle state involved in KdpFABC inhibition (*Figure 6*).

The catalytic states identified here closely adhere to the classical Post-Albers cycle of KdpFABC (*Video 1*). In the outward-facing E1 apo open state, the nucleotide-binding domain is widely accessible, and the N domain shows a high degree of flexibility, which likely enhances the efficiency of nucleotide binding from the environment. Once a nucleotide is bound, the N domain reorients toward the P domain for shared coordination of the nucleotide in the E1·ATP$_{early}$ state (*Figure 6*, transition 1a). This conformation shows a slightly more open nucleotide-bound state than the previously reported AMPPCP-stabilized E1·ATP state, which likely can progress closer to ATP cleavage (*Silberberg et al., 2021*; *Sweet*

**Figure 6.** Proposed transport cycle of KdpFABC in the absence and presence of regulatory phosphorylation at KdpB$_{S162}$. Black arrows indicate the non-inhibited catalytic cycle, while red arrows denote the transitions taken when KdpB$_{S162}$ is phosphorylated, and the complex is inhibited. Color code is as follows: KdpA in green, KdpC in purple, KdpB transmembrane domain (TMD) in sand with P domain in blue, N domain in red, and A domain in yellow. ATP shown as cyan pentagon; ADP as cyan rectangle; phosphorylated KdpB$_{D307}$ as cyan circle; K$^+$ ions in dark purple; the position of the TGES$_{162}$ loop in the A domain as an asterisk; KdpF is removed for simplicity. All published structures for each state are listed with PDB accession code and nucleotide conditions, with those from this work highlighted in bold.

*et al., 2021*). Subsequently, ATP cleavage leads to the phosphorylation of the catalytic KdpB$_{D307}$ in the P domain, forming the E1P·ADP intermediate (*Figure 6*, transition 2), which could be structurally isolated in this study under turnover conditions. The accumulation of the E1P·ADP state in both turnover samples analyzed in this study suggests that ADP and Mg$^{2+}$ release (*Figure 6*, transition 3) is the rate-limiting step of KdpFABC turnover. Following the catalytic cycle, the high-energy E1P state progresses to the E2P state, whereby large rearrangements bring the conserved TGES$_{162}$ loop in the A domain near the catalytic site of the P domain. This transition is accompanied by conformational changes in the TMD of KdpB, which switches the complex from an outward- to an inward-open state with respect to the CBS (*Figure 6*, transition 4a). K$^+$ is released to the cytosol and KdpB$_{D307-P}$ is dephosphorylated by the TGES$_{162}$ loop (*Figure 6*, transition 5). Subsequently, the complex cycles back to the E1 apo open state (*Figure 6*, transition 6).

Whereas the structures obtained for KdpFAB$_{S162-P}$C in the E1 apo open, E1·ATP, E1P·ADP, and E2P states align well with corresponding states in the Post-Albers cycle of other P-type ATPases, the nucleotide-free E1 apo tight and E1P tight states do not. Their overall compact fold is facilitated by KdpB$_{S162-P}$, which increases the contacts of the N and A domains. While the compact fold itself is not the main cause of stalling KdpFABC, it might contribute to stabilizing the inhibited complex limiting the innate flexibility of the N domain. Notably, the E1 apo tight state likely has no major physiological relevance, as in the presence of ATP, it would progress through the catalytic cycle up to the point of inhibition.

By contrast, we propose that the E1P tight state is involved in stalling the KdpFABC complex. We suggest that this state represents the biochemically described E1P inhibited state (*Sweet et al., 2020*), and is adopted after ADP release from the high-energy E1P state, when KdpFABC attempts to relax into the E2P state (*Figure 6*, transition 4b). This attempt would explain the partial tilt of the A domain in helix 4, observed only in the E1P tight structure. As suggested before (*Sweet et al., 2020*), the full relaxation to the E2P state is however hindered in the inhibited KdpFAB$_{S162-P}$C by a repulsion between the catalytic phosphate in the P domain (KdpB$_{D307-P}$) and the inhibitory phosphate in the A domain (KdpB$_{S162-P}$), which would come in close proximity during the transition from E1P to E2P (*Figure 6*, transition 4a, *Video 1*). The adopted stalled E1P tight state is likely an energetically favored state between E1P and E2P, easing the high-energy constraints around the phosphorylated KdpB$_{D307}$. In good agreement with this hypothesis, KdpFAB$_{S162-P}$C was stabilized in the E1P tight state even in the presence of the inhibitor orthovanadate, known to otherwise stabilize an E2P state. Notably, orthovanadate has the same charge distribution as the E1P tight state with phosphorylated KdpB$_{D307}$. By contrast, KdpFAB$_{S162-P}$C in the presence of AlF$_4^-$, which has a lower negative charge, could previously be stabilized in an E2P state (*Stock et al., 2018*), likely because the electrostatic repulsion is lower, making the E1P/E2P transition more favorable.

For a long time, E1P states of P-type ATPases have been biochemically characterized as high-energy intermediates. However, the structural determinants for this energetic unfavorability have yet to be described. The conformational changes we observe between the E1P·ADP state and the relaxed E1P tight state offer a first clue as to the regions involved in destabilizing the E1P state and triggering the transition to the E2P state. In the E1P tight state, the electrostatic tensions of the E1P·ADP state are eased by rearrangements of the catalytic aspartate and the surrounding side chains, likely supported by a pulling of the N domain on the D$_{307}$KTG loop as it associates with the A domain during the E1P to E1P tight transition. This could be the main stabilizing role of the tight state in KdpB$_{S162-P}$ inhibition. Further, the structural differences in the E1P tight state also involve a movement of helix 6 in the P domain, which is not in the immediate proximity of the catalytic KdpB$_{D307}$ phosphate. This helix is in close proximity to the A domain and next one of the two connections of the P domain to the TMD, which undergo large conformational changes during the E1/E2 transition. Thus, it may be an important element in signaling the ATPase to initiate the transition to E2P in the TMD from an outward- to an inward-facing state.

The regulation by inactive states outside the Post-Albers cycle has been postulated for other P-type ATPases (*Dyla et al., 2020*). Observation of translocation by the H$^+$ ATPase AHA2 on a single-molecule level revealed that the pump stochastically enters inactive states, from which it can return to its active form spontaneously (*Veshaguri et al., 2016*). These states are adopted from the E1 state, similar to the E1 apo tight state observed under nucleotide-free conditions for KdpFABC. However, no full inhibition in the E1P state was observed for AHA2, indicating a different mechanism. A crystal

structure of the Ca$^{2+}$ pump SERCA was also proposed to represent a non-Post-Albers state (*Dyla et al., 2020*; *Toyoshima et al., 2000*). Yet, the organization of the cytosolic domains differs significantly from the tight state seen for KdpB, and a physiological role for this structure remains unclear. Altogether, the structural basis for KdpFABC inhibition by KdpB$_{S162}$ phosphorylation presented here describes in detail the involvement of a non-Post-Albers state with a clear regulatory role in P-type ATPase turnover. It extends the conformational landscape and strongly supports the emerging idea of non-catalytic off-cycle states with important physiological roles.

## Conclusion

The data presented here illuminate the structural basis for KdpFABC inhibition by KdpB$_{S162}$ phosphorylation. We show that stalled KdpFABC adopts a novel conformation, which is either an intermediate of the E1P/E2P transition, or a separate state into which the A and P domains relax when the transition to the E2P state is hindered. Moreover, the N domain in the inhibited KdpFAB$_{S162-P}$C associates closely with the A domain in a non-Post-Albers state that likely further stabilizes the complex during inhibition. These results prove the involvement of off-cycle states in P-type ATPase regulation, and strongly support the burgeoning discussion of non-Post-Albers states of physiological relevance in the conformational landscapes of ion pumps. Further studies will be required to resolve how the phosphorylation of KdpB$_{S162}$ is mediated, and fully illuminate the destabilization of the E1P state and subsequent transition to the E2P state.

## Materials and methods

Key materials and software used in this work are listed in a key resources table.

**Key resources table**

| Reagent type (species) or resource | Designation | Source or reference | Identifiers | Additional information |
|---|---|---|---|---|
| Gene (*Escherichia coli*) | *kdpFABC* | | Uniprot IDs: *kdpF*: P36937; *kdpA*: P03959; *kdpB*: P03960; *kdpC*: P03961 | |
| Genetic reagent | *pBXC3H* | doi:10.1007/978-1-62703-764-8_11 | Addgene catalog #47068 | Plasmid backbone |
| Strain, strain background (*Escherichia coli*) | LB2003 (F⁻ *thi metE rpsL gal rha kup1 ΔkdpFABC5 ΔtrkA*) | doi:10.1007/s002030050425 | | |
| Sequence-based reagent | *kdpB_S162A_for* | Eurofins Genomics (Luxembourg) | | 5'-GCGCCATCACCGGGGAAGCGGCACC-3' |
| Sequence-based reagent | *kdpB_S162A_for* | Eurofins Genomics (Luxembourg) | | 5'-ACCGGTGCCGCTTCCCCGGTGATGG-3' |
| Sequence-based reagent | *kdpB_D307N_for* | Eurofins Genomics (Luxembourg) | | 5'-GCTACTGAATAAAACCGGCACCATCAC-3' |
| Sequence-based reagent | *kdpB_D307N_rev* | Eurofins Genomics (Luxembourg) | | 5'-GGTGCCGGTTTTATTCAGTAGCAGAACG-3' |
| Sequence-based reagent | *kdpB_A407C_for* | Eurofins Genomics (Luxembourg) | | 5'-CATTCGTCGCCATGTTGAGTGTAACGG-3' |
| Sequence-based reagent | *kdpB_A407C_rev* | Eurofins Genomics (Luxembourg) | | 5'-CGTTACACTCAACATGGCGACGAATGG-3' |
| Sequence-based reagent | *kdpB_A494C_for* | Eurofins Genomics (Luxembourg) | | 5'-GATTTTCTCGCCGAATGTACACCGGAGGCC-3' |
| Sequence-based reagent | *kdpB_A494C_rev* | Eurofins Genomics (Luxembourg) | | 5'-GGCCTCCGGTGTACATTCGGCGAGAAAATC-3' |

*Continued on next page*

*Continued*

| Reagent type (species) or resource | Designation | Source or reference | Identifiers | Additional information |
|---|---|---|---|---|
| Chemical compound, drug | MTSSL | Toronto Research Chemicals Inc, North York, Canada | | EPR spin label |
| Chemical compound, drug | Orthovanadate | Merck KGaA, Darmstadt, Germany | | |
| Chemical compound, drug | Adenosine 5'-triphosphate | Merck KGaA, Darmstadt, Germany | | |
| Software, algorithm | DeerAnalysis | doi:10.1007/BF03166213 | | |
| Other | 200 and 300 mesh Au R1.2/1.3 cryo-EM grids | Quantifoil, Großlöbichau, Germany | | Cryo-EM Grids |
| Software, algorithm | EPU v 2.3 | Thermo Fisher (Eindhoven, Netherlands) https://www.thermofisher.com/nl/en/home/electron-microscopy/products/software-em-3d-vis/epu-software.html | | |
| Software, algorithm | Sample thickness measurement script | https://doi.org/10.1101/2020.12.01.392100 | | |
| Software, algorithm | Focus | https://doi.org/10.1016/j.jsb.2017.03.007 https://focus.c-cina.unibas.ch/about.php | | |
| Software, algorithm | SBGrid | https://doi.org/10.7554/eLife.01456 https://sbgrid.org/software/ | | |
| Software, algorithm | MotionCor2 | https://doi.org/10.1038/nmeth.4193 http://msg.ucsf.edu/em/software/motioncor2.html | | |
| Software, algorithm | Ctffind 4.1.13 | https://doi.org/10.1016/j.jsb.2015.08.008 http://grigorriefflab.janelia.org/ctf | | |
| Software, algorithm | Ctffind 4.1.14 | https://doi.org/10.1016/j.jsb.2015.08.008 http://grigorriefflab.janelia.org/ctf | | |
| Software, algorithm | crYOLO 1.3.1 | https://doi.org/10.1038/s42003-019-0437-z https://cryolo.readthedocs.io/en/stable/ | | |
| Software, algorithm | crYOLO 1.7.6 | https://doi.org/10.1038/s42003-019-0437-z https://cryolo.readthedocs.io/en/stable/ | | |
| Software, algorithm | cryoSPARC 3.2.0 | https://doi.org/10.1038/nmeth.4169 https://cryosparc.com | | |
| Software, algorithm | RELION 3.0.8 | https://doi.org/10.7554/eLife.42166 https://www2.mrc-lmb.cam.ac.uk/relion/ | | |
| Software, algorithm | RELION 3.1.1 | https://doi.org/10.7554/eLife.42166 https://www2.mrc-lmb.cam.ac.uk/relion/ | | |
| Software, algorithm | Coot 0.9 | https://doi.org/10.1107/S0907444904019158 https://www2.mrc-lmb.cam.ac.uk/personal/pemsley/coot/ | | |

*Continued*

| Reagent type (species) or resource | Designation | Source or reference | Identifiers | Additional information |
|---|---|---|---|---|
| Software, algorithm | Phenix 1.19.2 | https://doi.org/10.1107/S2059798318006551 http://phenix-online.org/ | | |
| Software, algorithm | Molprobity 4.5.1 | https://doi.org/10.1002/pro.3330 http://molprobity.biochem.duke.edu | | |
| Software, algorithm | UCSF Chimera 1.15 | https://doi.org/10.1002/jcc.20084 https://www.cgl.ucsf.edu/chimera/ | | |
| Software, algorithm | UCSF Chimerax 1.1.1 | https://doi.org/10.1002/pro.3943 https://www.rbvi.ucsf.edu/chimerax/ | | |
| Software, algorithm | Origin Pro 2016 | OriginLab https://www.originlab.com/2016 | | |
| Software, algorithm | Gromacs 2019.4 | https://doi.org/10.1016/0010-4655(95)00042-E https://www.gromacs.org/ | | |
| Software, algorithm | VMD 1.9.4a12 | https://doi.org/10.1016/0263-7855(96)00018-5 http://www.ks.uiuc.edu/Research/vmd/ | | |
| Software, algorithm | charmm-gui | https://doi.org/10.1021/acs.jctc.5b00935 http://www.charmm-gui.org/ | | |
| Software, algorithm | MDAnalysis 1.0.1 | https://doi.org/10.1002/jcc.21787 https://www.mdanalysis.org/ | | |
| Software, algorithm | NumPy 1.20.0 | https://doi.org/10.1038/s41586-020-2649-2 https://numpy.org/ | | |
| Software, algorithm | matplotlib 3.3.4 | https://doi.org/10.1109/MCSE.2007.55 https://matplotlib.org/ | | |
| Software, algorithm | GraphPad Prism 8 and 9 | https://www.graphpad.com/scientific-software/prism/ | | |
| Software, algorithm | HOLE 2.2.004 | https://doi.org/10.1016/S0263-7855(97)00,009X | | |
| Software, algorithm | HOLLOW 1.3 | https://doi.org/10.1186/1472-6807-8-49 | | |

## Cloning, protein production, and purification

*Escherichia coli kdpFABC* (UniProt IDs: P36937 (KdpF), P03959 (KdpA), P03960 (KdpB), and P03961 (KdpC)) and its cysteine-free variant (provided by J.C. Greie, Osnabrück, Germany) were cloned into FX-cloning vector pBXC3H (pBXC3H was a gift from Raimund Dutzler & Eric Geertsma (*Geertsma and Dutzler, 2011*) (Addgene plasmid # 47068)) resulting in pBXC3H-KdpFABC and pBXC3H-KdpFABCΔCys, respectively. pBXC3H-KdpFAB$_{S162A}$C and pBXC3H-KdpFAB$_{D307N}$C were created from pBXC3H-KdpFABC by site-directed mutagenesis. Plasmids encoding variants used in EPR experiments include pBXC3H-KdpFAB$_{D307N/A407C/A494C}$CΔCys and pBXC3H-KdpFAB$_{S162A/D307N/A407C/A494C}$CΔCys, and were created by site-directed mutagenesis based on pBXC3H-KdpFABCΔCys.

KdpFABC and KdpFABC variants for structural analysis and pulsed EPR measurements were produced in *E. coli* LB2003 cells (available from the Hänelt group upon request) transformed with the respective plasmids in 12 l of KML (100 µg/ml ampicillin). Cell growth and harvesting were carried out as described previously (*Stock et al., 2018*). In brief, protein production was induced at an OD$_{600}$ of 1.0 by the addition of 0.002% L-arabinose for 1 hr at 37°C, after which cells were harvested at 5500 × *g* and 4°C. KdpFABC and KdpFABC variants were purified as previously described for wild-type

KdpFABC (*Stock et al., 2018*). In brief, cells in 50 mM Tris–HCl pH 7.5, 10 mM $MgCl_2$, 1 mM dithiothreitol (DTT), 10% glycerol, 2 mM Ethylenediaminetetraacetic acid (EDTA), and 0.5 mM phenylmethylsulfonyl fluoride (PMSF) were disrupted at 1 kbar via Stansted cell disruptor (Homogenising Systems Ltd, Essex, UK). Undisrupted cells were removed by centrifugation at 15,000 × *g* and membranes were harvested by centrifugation at 150,000 × *g* for 5 hr. Membranes in 50 mM Tris–HCl pH 7.5, 10 mM $MgCl_2$, 10% glycerol, and 0.5 mM PMSF were solubilized with 2% dodecyl maltoside (DDM) overnight, and unsolubilized membranes removed by centrifugation at 150,000 × *g* for 30 min. The solubilized fraction was supplemented with 150 mM NaCl and incubated with 1 ml pre-equilibrated $Ni^{2+}$-NTA for 1 hr at 4°C. Following removal of the flow-through, the column was washed with 50 column volumes wash buffer (50 mM Tris–HCl pH 7.5, 20 mM $MgCl_2$, 150 mM NaCl, 10% glycerol, 0.025% DDM) containing 30 mM imidazole, and then incubated with 1 CV wash buffer containing 1 mg/ml 3C protease for 1 hr. Elution fractions were pooled for further purification steps by AIEX (HiTrap Q HP column, Cytiva, Marlborough, MA, USA) with AIEX buffer (10 mM Tris–HCl pH 8, 10 mM $MgCl_2$, and 0.025% DDM) with a gradient from 10 to 150 mM NaCl and subsequent SEC (Superdex S200 Increase, GE Healthcare, Chicago, IL, USA) with SEC buffer (10 mM Tris–HCl pH 8, 10 mM $MgCl_2$, 10 mM NaCl, and 0.0125% DDM).

## Cryo-EM sample preparation

All cryo-EM samples of KdpFABC were prepared in 10 mM Tris–HCl pH 8.0, 10 mM $MgCl_2$, 10 mM NaCl, 0.0125% DDM, and were subsequently supplemented as described.

### Wild-type KdpFABC under turnover conditions

Purified wild-type KdpFABC was concentrated to 5 mg/ml and supplemented with 50 mM KCl and 2 mM ATP. The sample was incubated at 24°C for 5 min before grid preparation.

### Wild-type KdpFABC stabilized by orthovanadate

Purified wild-type KdpFABC was concentrated to 4 mg/ml and supplemented with 1 mM KCl and 0.2 mM orthovanadate before grid preparation. Orthovanadate was prepared by dissolving in water, adjusting the pH, and subsequent heating and pH adjustment iterations until no further discoloration was observable (*Csermely et al., 1985*).

### KdpFAB$_{S162A}$C under turnover conditions

Purified KdpFAB$_{S162A}$C, in which the inhibitory phosphorylation in KdpB is prevented by the mutation of the phosphorylated KdpB$_{S162}$ to alanine, was concentrated to 3.4 mg/ml and supplemented with 50 mM KCl and 2 mM ATP. The sample was incubated at 24°C for 5 min before grid preparation.

### KdpFAB$_{D307N}$C under nucleotide-free conditions

Purified KdpFAB$_{D307N}$C, which is prevented from progressing into an E1P state by the mutation of the catalytic KdpB$_{D307}$ to an asparagine, was concentrated to 4 mg/ml and supplemented with 50 mM KCl before grid preparation.

## Cryo-EM grid preparation

For wild-type KdpFABC with orthovanadate and nucleotide-free KdpFAB$_{D307N}$C, 2.8 μl of sample were applied to holey-carbon cryo-EM grids (Quantifoil Au R1.2/1.3, 200 mesh), which were previously glow-discharged at 5 mA for 20 s. Grids were blotted for 3–5 s in a Vitrobot (Mark IV, Thermo Fisher Scientific) at 20°C and 100% humidity, and subsequently plunge-frozen in liquid propane/ethane and stored in liquid nitrogen until further use.

For the turnover samples of wild-type KdpFABC and KdpFAB$_{S162A}$C, 2.8 μl of sample were applied to holey-carbon cryo-EM grids (Quantifoil Au R1.2/1.3, 300 mesh), which were previously twice glow-discharged at 15 mA for 45 s. Grids were blotted for 2–6 s in a Vitrobot (Mark IV, Thermo Fisher Scientific) at 4°C and 100% humidity, and subsequently plunge-frozen in liquid ethane and stored in liquid nitrogen until further use.

## Cryo-EM data collection

Cryo-EM data were collected on a 200 keV Talos Arctica microscope (Thermo Fisher Scientific) equipped with a post-column energy filter (Gatan) in zero-loss mode, using a 20 eV slit, a 100 μm objective aperture, in an automated fashion provided by EPU software (Thermo Fisher Scientific) or serialEM (*Mastronarde, 2005*; *Schorb et al., 2019*) on a K2 summit detector (Gatan) in counting mode. Cryo-EM images were acquired at a pixel size of 1.012 Å (calibrated magnification of ×49,407), a defocus range from −0.5 to −2 μm, an exposure time of 9 s and a subframe exposure time of 150 ms (60 frames), and a total electron exposure on the specimen level of about 52 electrons per $Å^2$. Data collection was optimized by restricting the acquisition to regions displaying optimal sample thickness using an in-house written script (*Rheinberger et al., 2021*) and data quality was monitored on-the-fly using the software FOCUS (*Biyani et al., 2017*).

## Cryo-EM data processing

For all datasets, the SBGrid (*Morin et al., 2013*) software package tool was used to manage the software packages.

### KdpFAB$_{S162A}$C under turnover conditions

A total of 11,482 dose-fractionated cryo-EM images were recorded and subjected to motion-correction and dose-weighting of frames by MotionCor2 (*Zheng et al., 2017*). The CTF parameters were estimated on the movie frames by ctffind4.1.4 (*Rohou and Grigorieff, 2015*). Bad images showing contamination, a defocus below −0.5 or above −2.0 μm, or a bad CTF estimation were discarded, resulting in 9170 images used for further analysis with the software packages cryoSPARC 3.2.0 (*Punjani et al., 2017*) and RELION 3.1.1 (*Zivanov et al., 2018*). First, crYOLO 1.7.6 (*Wagner et al., 2019*) was used to automatically pick 287,232 particles using a loose threshold. Particle coordinates were imported in RELION 3.1.1 (*Zivanov et al., 2018*), and the particles were extracted with a box size of 240 pixels. Non-protein classes were removed with a single round of 2D classification in cryoSPARC 3.2.0 (*Punjani et al., 2017*), resulting in 167,721 particles (initial particle set). These particles were then subjected to ab initio 3D reconstruction in cryoSPARC 3.2.0 (*Punjani et al., 2017*), and the best two output classes were used in subsequent jobs in an iterative way in RELION 3.1.1 (*Zivanov et al., 2018*). From here on the classes were treated separately, with about 37.7% (63,240 particles) in the E2P state and about 53.3% (89,378 particles) in the E1P·ADP state. These particles were imported back into RELION 3.1.1, and subjected to 3D classification and refinement, against references obtained for the E1 tight and E1P·ADP state. This resulted in a dataset of 46,904 particles (~28% of the initial particle set) for the E2P state, and of 70,068 particles for the E1P·ADP state. Several rounds of CTF refinement (*Zivanov et al., 2018*) were performed, using per-particle CTF estimation. The dataset for the E1P·ADP state was subjected to a round of focused 3D classification with no image alignment, using a mask on the flexible A and N domains of KdpB (*Hiraizumi et al., 2019*). This resulted in a cleaned dataset of 58,243 particles (~35% of the initial particle set) for the E1P·ADP state. In the last refinement iteration, a mask excluding the micelle was used and the refinement was continued until convergence (focused refinement), yielding a final map for the E2P state at a resolution of 4.3 after refinment and 4.0 Å after post-processing and masking, sharpened using an isotropic *b*-factor of −160 $Å^2$. The final map for the E1P·ADP state had a resolution of 4.0 Å after refinement and 3.7 Å after post-processing and masking, and was sharpened using an isotropic *b*-factor of −123 $Å^2$.

### Wild-type KdpFABC under turnover conditions

Pre-processing of the acquired data was performed as described above, resulting in the selection of 14,604 out of 17,938 images, which were used for further analysis with the software packages cryoSPARC 3.2.0 (*Punjani et al., 2017*) and RELION 3.1.1 (*Zivanov et al., 2018*). First, crYOLO 1.7.6 (*Wagner et al., 2019*) was used to automatically pick 1,128,433 particles using a loose threshold. Particle coordinates were imported in RELION 3.1.1 (*Zivanov et al., 2018*), and the particles were extracted with a box size of 240 pixels. Non-protein classes were removed with a single round of 2D classification in cryoSPARC 3.2.0 (*Punjani et al., 2017*), resulting in 828,847 particles (initial particle set). These particles were then subjected to ab initio 3D reconstruction in cryoSPARC 3.2.0 (*Punjani et al., 2017*), and the best three output classes were used in subsequent jobs in an iterative way in RELION 3.1.1 (*Zivanov et al., 2018*). From here on the classes were treated separately, with about

33.2% (275,026 particles) in the E1P tight state and about 55.1% (346,303 and 110,601 particles) in the E1 nucleotide-bound state. These particles were imported back into RELION 3.1.1, and subjected to 3D classification and refinement, against references obtained for the E1 tight and E1·ATP states. Several rounds of CTF refinement (*Zivanov et al., 2018*) were performed, using per-particle CTF estimation, before subjecting all datasets to a round of focused 3D classification with no image alignment, using a mask on the flexible A and N domains of KdpB (*Hiraizumi et al., 2019*). This resulted in a cleaned dataset of 114,588 particles (~14% of the initial particle set) for the E1P tight state, and 277,912 and 80,798 for the E1 nucleotide-bound states. The latter two were merged and subjected to several rounds of CTF refinement (*Zivanov et al., 2018*) using per-particle CTF estimation, before subjecting the dataset to another round of focused 3D classification with no image alignment, using a mask on the flexible A and N domains of KdpB (*Hiraizumi et al., 2019*). This resulted in two distinct datasets of 257,675 particles (~31% of the initial particle set) for the E1P·ADP state and of 76,121 particles (~9% of the initial particle set) for the E1·ATP state. In the last refinement iteration, a mask excluding the micelle was used and the refinement was continued until convergence (focused refinement), yielding a final map for the E1P tight state at a resolution of 3.7 after refinement and 3.4 Å after post-processing and masking, sharpened using an isotropic *b*-factor of −134 Å$^2$. The final map for the E1P·ADP state had a resolution of 3.4 Å after refinement and 3.1 Å after post-processing and masking, and was sharpened using an isotropic *b*-factor of −122 Å$^2$. The final map for the E1·ATP state had a resolution of 3.9 Å after refinement and 3.5 Å after post-processing and masking, and was sharpened using an isotropic *b*-factor of −132 Å$^2$. No symmetry was imposed during 3D classification or refinement.

## Wild-type KdpFABC with orthovanadate

Pre-processing of the acquired data was performed as described above, resulting in the selection of 2014 out of 2,488 images, which were used for further analysis with the software package RELION 3.0.8 (*Zivanov et al., 2018*). First, crYOLO 1.3.1 (*Wagner et al., 2019*) was used to automatically pick 164,891particles using a loose threshold. Particle coordinates were imported in RELION 3.0.8 (*Zivanov et al., 2018*), and the particles were extracted with a box size of 240 pixels. Non-protein classes were removed with 2D classification, resulting in 120,077 particles (initial particle set). For 3D classification and refinement, the map of the previously generated E1 state EMD-0257 (*Stock et al., 2018*) was used as reference for the first round, and the two best output classes were used in subsequent jobs in an iterative way. From here on the classes were treated separately, with about 70.9% (85,102 particles) in the E1P tight state and about 11.2% (13,508 particles) in the E2P state. Sequentially, on the E1P tight dataset several rounds of CTF refinement, using per-particle CTF estimation, and Bayesian polishing were performed (*Zivanov et al., 2018*), before subjecting the dataset to a round of focused 3D classification with no image alignment, using a mask on the flexible A and N domains of KdpB (*Hiraizumi et al., 2019*). This resulted in a cleaned dataset of 74,927 particles (~62% of the initial particle set) for the E1P tight state, and was subjected to several rounds of CTF refinement, using per-particle CTF estimation, and Bayesian polishing (*Zivanov et al., 2018*). In the last refinement iteration, a mask excluding the micelle was used and the refinement was continued until convergence (focused refinement), yielding a final map for the E1P tight state at a resolution of 3.3 Å after refinement and 3.3 Å after post-processing and masking, sharpened using an isotropic *b*-factor of −55 Å$^2$. The final map for the E2P state (from ~11% of the initial particle set) had a resolution of 8.7 Å after refinement and 7.4 Å after post-processing and masking, and was sharpened using an isotropic *b*-factor of −195 Å$^2$. No symmetry was imposed during 3D classification or refinement.

## KdpFAB$_{D307N}$C under nucleotide-free conditions

Pre-processing of the acquired data was performed as described above, resulting in the selection of 12,864 out of 17,889 images, which were used for further analysis with the software package RELION 3.0.8 (*Zivanov et al., 2018*). First, crYOLO 1.3.1 (*Wagner et al., 2019*) was used to automatically pick 728,674 particles using a loose threshold. Particle coordinates were imported in RELION 3.0.8 (*Zivanov et al., 2018*), and the particles were extracted with a box size of 240 pixels. Non-protein classes were removed in several rounds of 2D classification, resulting in 469,466 particles (initial particle set). Due to the large conformational differences between both states, the full dataset was further cleaned by two independent 3D classifications against references obtained for the E1 tight

state or the E1 apo open state. Particles belonging to the best classes of both runs were merged and duplicates subtracted, resulting in 306,942 particles that were subjected to a multi-reference 3D classification with no image alignment. From here on the classes were treated separately, with about 33.7% (158,353 particles) in the E1 tight state and about 31.4% (147,589 particles) in the open state. Several rounds of CTF refinement (*Zivanov et al., 2018*) were performed, using per-particle CTF estimation, before subjecting both datasets to a round of focused 3D classification with no image alignment, using a mask on the flexible A and N domains of KdpB (*Hiraizumi et al., 2019*). This resulted in a cleaned dataset of 88,852 particles (~19% of the initial particle set) for the E1 tight state, 75,711 particles (~16% of the initial particle set) for the E1 apo open state 1, and 47,981 particles (~10% of the initial particle set) for the E1 apo open state 2. In the last refinement iteration, a mask excluding the micelle was used and the refinement was continued until convergence (focused refinement), yielding a final map for the E1 tight state at a resolution of 3.8 after refinement and 3.4 Å after post-processing and masking, sharpened using an isotropic *b*-factor of −113 Å$^2$. The final map for the E1 apo open state 1 had a resolution of 3.9 Å after refinement and 3.5 Å after post-processing and masking, and was sharpened using an isotropic *b*-factor of −117 Å$^2$. The final map for the E1 apo open state 2 had a resolution of 4.0 Å after refinement and 3.7 Å after post-processing and masking, and was sharpened using an isotropic *b*-factor of −119 Å$^2$.

For all datasets, local resolution estimates were calculated by RELION and no symmetry was imposed during 3D classification or refinement. All resolutions were estimated using the 0.143 cutoff criterion (*Rosenthal and Henderson, 2003*) with gold-standard Fourier shell correlation (FSC) between two independently refined half-maps. During post-processing, the approach of high-resolution noise substitution was used to correct for convolution effects of real-space masking on the FSC curve (*Chen et al., 2013*).

## Model building and validation

Available KdpFABC structures like E1·ATP state [7NNL], E1 state [6HRA], and E2 state [6HRB] were docked into the obtained cryo-EM maps using UCSF Chimera (*Pettersen et al., 2004*) and used as initial models. Wherever required, rigid body movements were applied to accommodate for conformational changes, and models were subjected to an iterative process of real space refinement using Phenix.real_space_refinement with secondary structure restraints (*Afonine et al., 2018*; *Liebschner et al., 2019*) followed by manual inspection and adjustments in Coot (*Emsley and Cowtan, 2004*). K$^+$ ions, cardiolipin, ATP, ADP, P$_i$, and orthovanadate were modeled into the cryo-EM maps in Coot. The final models were refined in real space with Phenix.real_space_refinement with secondary structure restraints (*Afonine et al., 2018*; *Liebschner et al., 2019*). For validation of the refinement, FSCs (FSC$_{sum}$) between the refined models and the final maps were determined. To monitor the effects of potential over-fitting, random shifts (up to 0.5 Å) were introduced into the coordinates of the final model, followed by refinement against the first unfiltered half-map. The FSC between this shaken-refined model and the first half-map used during validation refinement is termed FSC$_{work}$, and the FSC against the second half-map, which was not used at any point during refinement, is termed FSC$_{free}$. The marginal gap between the curves describing FSC$_{work}$ and FSC$_{free}$ indicate no over-fitting of the model. The geometries of the atomic models were evaluated by MolProbity (*Williams et al., 2018*).

## RMSD calculations

For conformational comparisons, structures were superimposed in KdpA, which is rigid across all conformational states. RMSDs were calculated for the cytosolic domains of KdpB (residues 89–215, 275–569), which feature the largest conformational differences, using ChimeraX (*Pettersen et al., 2021*). Structural comparisons involving the crystal structure [5MRW] were performed using superpositions in KdpA (column 5MRW A) and in the TM domain of KdpB (residues 1–88, 216–275, 570–682; column 5MRW B) to compensate structural deviations in the TMD that skew the structural alignment of the KdpB subunit.

## Tunnel calculations

Pore calculations in KdpA were performed with the software HOLE (*Smart et al., 1996*). For this, a Conolly probe of radius 0.9 Å was used on a PDB of KdpA in which cofactors and other subunits were removed and the translocation pore was aligned parallel to the *z*-axis.

The tunnel from KdpA to KdpB and the inward-open half-channel from the KdpB CBS to the cytosol were calculated with HOLLOW, using a probe radius of 1.2 Å and a starting point immediately below the KdpA SF or in the KdpB CBS, respectively (*Ho and Gruswitz, 2008*).

## ATPase assay

The ATPase activity of purified KdpFABC variants was tested using a malachite green-based ATPase assay (*Carter and Karl, 1982*). Each reaction contained 2 mM ATP and 1 mM KCl, and was started by adding 0.5–1.0 μg protein. The reaction was carried out for 5 min at 37°C.

## EPR sample purification, preparation, data acquisition and analysis

KdpFAB$_{A407C/A494C}$CΔCys, KdpFAB$_{D307N/A407C/A494C}$CΔCys, and KdpFAB$_{S162A/D307N/A407C/A494C}$CΔCys, variants based on an otherwise Cys-less background, were produced and purified as described for KdpFAB$_{G150C/A407C}$CΔCys (*Stock et al., 2018*). Purified and spin-labeled KdpFABCΔCys variants were concentrated to 4–7 mg ml$^{-1}$ and supplemented with 14% deuterated glycerol (vol/vol) and 50 mM KCl. 5 mM AMPPCP stabilizing the E1-ATP states was added when indicated.

Pulsed EPR measurements were performed at Q band (34 GHz) and −223°C on an ELEXSYS-E580 spectrometer (Bruker). For this, 15 μl of the freshly prepared samples were loaded into EPR quartz tubes with a 1.6-mm outer diameter and shock frozen in liquid nitrogen. During the measurements, the temperature was controlled by the combination of a continuous-flow helium cryostat (Oxford Instruments) and a temperature controller (Oxford Instruments). The four-pulse DEER sequence was applied (*Pannier et al., 2011*) with observer pulses of 32 ns and a pump pulse of 12–14 ns. The frequency separation was set to 70 MHz and the frequency of the pump pulse to the maximum of the nitroxide EPR spectrum. Validation of the distance distributions was performed by means of the validation tool included in DeerAnalysis (*Jeschke et al., 2006*) and varying the parameters 'Background start' and 'Background density' in the suggested range by applying fine grid. A prune level of 1.15 was used to exclude poor fits. Furthermore, interspin distance predictions were carried out by using the rotamer library approach included in the MMM software package (*Jeschke, 2018*; *Polyhach et al., 2011*). The calculation of the interspin distance predictions is based on the cryo-EM structures of the E1 tight, E1 apo open, and E1·ATP states for the comparison with the experimentally determined interspin distance distributions. To quantify DEER EPR signals, the distance distributions were approximated with Gaussian regression, and the areas under each curve were used to quantify the abundance of each state.

## MD simulations

MD simulations were built using the coordinates of eight states of the complex (see *Table 3*). To reduce the size of the simulation box, KdpA and KdpC were removed from the system, as these were considered unlikely to impact the dynamics of the N, P, and A domains in the timescales simulated. The systems were described with the CHARMM36m force field (*Best et al., 2012*; *Huang et al., 2017b*) and built into POPE membranes with TIP3P waters and K$^+$ and Cl$^-$ to 150 mM, using CHARMM-GUI (*Jo et al., 2007*; *Lee et al., 2016*). Where present, KdpB$_{S162}$ was phosphorylated for each system, and for the E1·ATP and E1P·ADP states, the nucleotide was included based on the structural coordinates. Where present, the K$^+$ bound in the CBS was preserved.

Each system was minimized using the steepest descents method, then equilibrated with positional restraints on heavy atoms for 100 ps in the NPT ensemble at 310 K with the V-rescale thermostat and a 1 ps coupling time constant, and semi-isotropic Parrinello–Rahman pressure coupling at 1 atm, with a 5-ps coupling time constant (*Bussi*

**Table 3.** Details of molecular dynamics (MD) simulations run.

All simulations were run in POPE membranes, over 3 × 50 ns. Root mean square deviations (RMSDs) are the mean and standard deviations over three repeats.

| State | RMSD (nm) |
|---|---|
| E1 apo open 1 (nucleotide-free) | 0.47 ± 0.09 |
| E1 apo open 2 (nucleotide-free) | 0.55 ± 0.12 |
| E1 apo tight (nucleotide-free) | 0.32 ± 0.08 |
| E1·ATP$_{early}$ (turnover WT) | 0.35 ± 0.04 |
| E1P·ADP (turnover WT) | 0.51 ± 0.04 |
| E1P·ADP (turnover KdpB$_{S162A}$) | 0.30 ± 0.06 |
| E1P tight (turnover WT) | 0.31 ± 0.04 |
| E1P tight (orthovanadate) | 0.27 ± 0.03 |

**Table 4.** EMDB and PDB accession codes of structure depositions.

| State | EMDB # | PDB # | EMPIAR # |
|---|---|---|---|
| E1·ATP$_{early}$ (turnover WT) | EMD-14913 | 7ZRG | EMPIAR-11232 |
| E1P·ADP (turnover WT) | EMD-14917 | 7ZRK | EMPIAR-11232 |
| E1P tight (turnover WT) | EMD-14912 | 7ZRE | EMPIAR-11232 |
| E1P tight (orthovanadate) | EMD-14911 | 7ZRD | EMPIAR-11230 |
| E2P (orthovanadate) | EMD-14347 | N/A | EMPIAR-11230 |
| E1P·ADP (turnover KdpB$_{S162A}$) | EMD-14919 | 7ZRM | EMPIAR-11231 |
| E2P (turnover KdpB$_{S162A}$) | EMD-14918 | 7ZRL | EMPIAR-11231 |
| E1 apo tight (nucleotide-free KdpB$_{D307N}$) | EMD-14914 | 7ZRH | EMPIAR-11229 |
| E1 apo open 1 (nucleotide-free KdpB$_{D307N}$) | EMD-14915 | 7ZRI | EMPIAR-11229 |
| E1 apo open 2 (nucleotide-free KdpB$_{D307N}$) | EMD-14916 | 7ZRJ | EMPIAR-11229 |

*et al., 2007*; *Parrinello and Rahman, 1981*). Production simulations were run using 2 fs time steps over 50 ns, with three repeats run for each state. The simulations were kept relatively short to preserve the conformation of the input structures, whilst allowing sufficient conformational flexibility to sample the side-chain motions and rearrangements within the given state. Removal of KdpA and KdpC did not appear to reduce the stability of KdpBF, as all systems had moderate–low backbone RMSDs at the end of the simulations (see *Table 3*).

Contact analysis was performed by counting the number of residues from each domain which were within 0.4 nm of a residue from a different domain, for every frame over 3 × 50 ns simulation. The domains were defined as the following residues of KdpB: A domain = 89–214, N domain = 314–450, and P-domain = 277–313 and 451–567. Contact analysis was run with the Gromacs tool gmx select.

High-frequency contacting residue pairs were identified as any pair of residues in contact for at least 90% of frames over 3 × 50 ns of simulation time. Analyses were run using MDAnalysis (*Michaud-Agrawal et al., 2011*) and plotted using NumPy (*Harris et al., 2020*) and Matplotlib (*Hunter, 2007*).

All simulations were run in Gromacs 2019 (*Berendsen et al., 1995*).

## Figure preparation

All figures were prepared using USCF Chimera (*Pettersen et al., 2004*), UCSF ChimeraX (*Pettersen et al., 2021*), VMD (*Humphrey et al., 1996*), OriginPro 2016, and GraphPad Prism 8 and 9.

## Data availability

The three-dimensional cryo-EM densities, the raw data and corresponding modeled coordinates of KdpFABC have been deposited in the Electron Microscopy Data Bank (EMDB), the Electron Microscopy Public Image Archive (EMPIAR), and the Protein Data Bank (PDB) under the accession numbers summarized in *Table 4*. The depositions include maps calculated with higher *b*-factors, both half-maps and the mask used for the final FSC calculation.

## Acknowledgements

The authors thank Werner Kühlbrandt and Sonja Welsch for the use of their vitrobot for freezing of turnover cryo-EM samples. CP thanks Michiel Punter for IT support. JMS thanks Janina Stautz for support with cryo-EM sample freezing. JMS thanks Paul JN Böhm for assistance with cloning and cell growth. PJS acknowledges the University of Warwick Scientific Computing Research Technology Platform for computational access. The electron microscopy within this work is part of the research program National Roadmap for Large-Scale Research Infrastructure (NEMI), project number 184.034.014, which is financed by the Dutch Research Council (NWO).

# Additional information

## Funding

| Funder | Grant reference number | Author |
|---|---|---|
| Nederlandse Organisatie voor Wetenschappelijk Onderzoek | Veni grant 722.017.001 | Cristina Paulino |
| Nederlandse Organisatie voor Wetenschappelijk Onderzoek | Start-Up grant 740.018.016 | Cristina Paulino |
| Deutsche Forschungsgemeinschaft | Emmy Noether grant HA6322/3-1 | Inga Hänelt |
| Deutsche Forschungsgemeinschaft | Heisenberg program HA6322/5-1 | Inga Hänelt |
| Aventis Foundation | Life Science Bridge Award | Inga Hänelt |
| Uniscientia Foundation | | Inga Hänelt |
| Wellcome Trust | 208361/Z/17/Z | Phillip J Stansfeld Robin A Corey |
| Medical Research Council | MR/S009213/1 | Phillip J Stansfeld |
| Biotechnology and Biological Sciences Research Council | BB/P01948X/1 | Phillip J Stansfeld |
| Biotechnology and Biological Sciences Research Council | BB/R002517/1 | Phillip J Stansfeld |
| Biotechnology and Biological Sciences Research Council | BB/S003339/1 | Phillip J Stansfeld |
| State of Hesse | LOEWE Schwerpunkt TRABITA | Jakob M Silberberg |

The funders had no role in study design, data collection, and interpretation, or the decision to submit the work for publication. For the purpose of Open Access, the authors have applied a CC BY public copyright license to any Author Accepted Manuscript version arising from this submission.

## Author contributions

Jakob M Silberberg, Charlott Stock, Lisa Hielkema, Robin A Corey, Formal analysis, Validation, Investigation, Visualization, Writing – original draft, Writing – review and editing; Jan Rheinberger, Dorith Wunnicke, Formal analysis, Validation, Investigation; Victor RA Dubach, Formal analysis, Investigation; Phillip J Stansfeld, Inga Hänelt, Cristina Paulino, Conceptualization, Supervision, Funding acquisition, Project administration, Writing – review and editing

## Author ORCIDs

Jakob M Silberberg ⬥ http://orcid.org/0000-0003-1721-8666
Charlott Stock ⬥ http://orcid.org/0000-0001-5471-3696
Robin A Corey ⬥ http://orcid.org/0000-0003-1820-7993
Jan Rheinberger ⬥ http://orcid.org/0000-0002-9901-2065
Victor RA Dubach ⬥ http://orcid.org/0000-0002-1657-7184
Inga Hänelt ⬥ http://orcid.org/0000-0003-1495-3163
Cristina Paulino ⬥ http://orcid.org/0000-0001-7017-109X

## Decision letter and Author response

Decision letter https://doi.org/10.7554/eLife.80988.sa1
Author response https://doi.org/10.7554/eLife.80988.sa2

# Additional files

## Supplementary files
• MDAR checklist

## Data availability

The three-dimensional cryo-EM densities and corresponding modeled coordinates generated have been deposited in the Electron Microscopy Data Bank and the Protein Data Bank under the accession numbers summarized in Table 4. The depositions include maps calculated with higher *b*-factors, both half-maps and the mask used for the final FSC calculation.

The following datasets were generated:

| Author(s) | Year | Dataset title | Dataset URL | Database and Identifier |
|---|---|---|---|---|
| Silberberg JM | 2022 | KdpFABC WT (KdpB-Ser162-P) in an E1·ATPearly state under turnover conditions | https://www.ebi.ac.uk/pdbe/entry/pdb/7zrg | Electron Microscopy Data Bank, 7zrg |
| Silberberg JM | 2022 | KdpFABC WT (KdpB-Ser162-P) in an E1P·ADP state under turnover conditions | https://www.ebi.ac.uk/pdbe/entry/pdb/7zrk | Electron Microscopy Data Bank, 7zrk |
| Silberberg JM | 2022 | KdpFABC WT (KdpB-Ser162-P) in an E1P-tight state under turnover conditions | https://www.ebi.ac.uk/pdbe/entry/pdb/7zre | Electron Microscopy Data Bank, 7zre |
| Silberberg JM | 2022 | KdpFABC WT (KdpB-Ser162-P) in an E1P-tight state in presence of orthovanadate | https://www.ebi.ac.uk/pdbe/entry/pdb/7zrd | Electron Microscopy Data Bank, 7zrd |
| Silberberg JM | 2022 | KdpFAB(Ser162-P,D307N)C in an E1 apo tight state | https://www.ebi.ac.uk/pdbe/entry/pdb/7zrh | Electron Microscopy Data Bank, 7zrh |
| Silberberg JM | 2022 | KdpFAB(Ser162-P,D307N)C in an E1 apo open 1 state | https://www.ebi.ac.uk/pdbe/entry/pdb/7zri | Electron Microscopy Data Bank, 7zri |
| Silberberg JM | 2022 | KdpFAB(Ser162-P,D307N)C in an E1 apo open 2 state | https://www.ebi.ac.uk/pdbe/entry/pdb/7zrj | Electron Microscopy Data Bank, 7zrj |
| Silberberg JM | 2022 | KdpFAB(S162A) in an E1P·ADP state under turnover conditions | https://www.ebi.ac.uk/pdbe/entry/pdb/7zrm | Electron Microscopy Data Bank, 7zrm |
| Silberberg JM | 2022 | KdpFABC WT (KdpB-Ser162-P) in an E2P state under turnover conditions | https://www.ebi.ac.uk/pdbe/entry/pdb/7zrl | Electron Microscopy Data Bank, 7zrl |
| Silberberg JM | 2022 | KdpFABC WT (KdpB-Ser162-P) in an E2P state in presence of orthovanadate | https://www.ebi.ac.uk/emdb/EMD-14347 | Electron Microscopy Data Bank, EMD-14347 |
| Silberberg JM | 2022 | Single particle cryo-EM of KdpFABC WT (KdpB-Ser162-P) under turnover condition | https://www.ebi.ac.uk/empiar/EMPIAR-11232/ | Electron Microscopy Public Image Archive, EMPIAR-11232 |
| Silberberg JM | 2022 | Single particle cryo-EM of KdpFAB(S162A)C under turnover condition | https://www.ebi.ac.uk/empiar/EMPIAR-11231/ | Electron Microscopy Public Image Archive, EMPIAR-11231 |
| Silberberg JM | 2022 | Single particle cryo-EM of KdpFABC WT (KDpB-Ser162-P) in presence of orthovanadate | https://www.ebi.ac.uk/empiar/EMPIAR-11230/ | Electron Microscopy Public Image Archive, EMPIAR-11230 |

*Continued*

| Author(s) | Year | Dataset title | Dataset URL | Database and Identifier |
|---|---|---|---|---|
| Silberberg JM | 2022 | Single particle cryo-EM of KdpFAB(Ser162-P, D307N)C in apo condition | https://www.ebi.ac.uk/empiar/EMPIAR-11229/ | Electron Microscopy Public Image Archive, EMPIAR-11229 |
| Silberberg JM | 2022 | KdpFABC WT (KdpB-Ser162-P) in an E1·ATPearly state under turnover conditions | https://www.ebi.ac.uk/emdb/EMD-14913 | Electron Microscopy Data Bank, EMDB-14913 |
| Silberberg JM | 2022 | KdpFABC WT (KdpB-Ser162-P) in an E1P·ADP state under turnover conditions | https://www.ebi.ac.uk/emdb/EMD-14917 | Electron Microscopy Data Bank, EMD-14917 |
| Silberberg JM | 2022 | KdpFABC WT (KdpB-Ser162-P) in an E1P-tight state under turnover conditions | https://www.ebi.ac.uk/emdb/EMD-14912 | Electron Microscopy Data Bank, EMD-14912 |
| Silberberg JM | 2022 | KdpFABC WT (KdpB-Ser162-P) in an E1P-tight state in presence of orthovanadate | https://www.ebi.ac.uk/emdb/EMD-14911 | Electron Microscopy Data Bank, EMD-14911 |
| Silberberg JM | 2022 | KdpFAB(Ser162-P,D307N)C in an E1 apo tight state | https://www.ebi.ac.uk/emdb/EMD-14914 | Electron Microscopy Data Bank, EMD-14914 |
| Silberberg JM | 2022 | KdpFAB(Ser162-P,D307N)C in an E1 apo open 1 state | https://www.ebi.ac.uk/emdb/EMD-14915 | Electron Microscopy Data Bank, EMD-14915 |
| Silberberg JM | 2022 | KdpFAB(Ser162-P,D307N)C in an E1 apo open 2 state | https://www.ebi.ac.uk/emdb/EMD-14916 | Electron Microscopy Data Bank, EMD-14916 |
| Silberberg JM | 2022 | KdpFAB(S162A) in an E1P·ADP state under turnover conditions | https://www.ebi.ac.uk/emdb/EMD-14919 | Electron Microscopy Data Bank, EMD-14919 |
| Silberberg JM | 2022 | KdpFABC WT (KdpB-Ser162-P) in an E2P state under turnover conditions | https://www.ebi.ac.uk/emdb/EMD-14918 | Electron Microscopy Data Bank, EMD-14918 |

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
