## [Editor Report]

KdpFABC is a bacterial potassium uptake transporter made up of a channel-like subunit (KdpA) and a P-type ATPase (KdpB). When potassium levels are low (< 2 mM), the transporter actively and selectively uptakes potassium, but must be switched off again to prevent excessive potassium accumulation. Here, by using cryo-EM, pulsed EPR measurements and MD simulations the molecular basis KdpFABC for inhibition by phosphorylation has been defined to an arrested (off-state) that is in a distinct conformation from previously determined P-type ATPase structures.

---

## [Decision Letter]

**Decision letter after peer review:**

Thank you for submitting your article "Inhibited KdpFABC resides in an E1 off-cycle state" for consideration by *eLife*. Your article has been reviewed by 3 peer reviewers, and the evaluation has been overseen by a Reviewing Editor and Kenton Swartz as the Senior Editor. The following individual involved in the review of your submission has agreed to reveal their identity: David Drew (Reviewer #1).

Essential revisions:

1. At the heart of this study is the comparison of the newly determined KdpFABC structures with previously published ones (of which there are already 10). Yet, an overview of the already determined structures is missing. There are also no RMSD calculations to illustrate the magnitude of any structural deviations, which makes it hard to judge the true level of novelty/deviation from known structures. Please revise and report the r.m.s.d. differences and superimposition and, where required, highlight structure features consistent with the novel states. This is of particular importance for the E1-P·ATP tight structure, which is reported as novel and yet it's unclear how similar this structure is to the previously published crystal structure (5MRW). There also seemed to be few changes in the TM segments that one would expect for inward and outward-facing states.

2. The paper suggests that KdpFABC cannot undergo the transition from the E1P tight to E2P and gets stuck in this dead-end 'off cycle' state. To test this, the authors analysed an S162-P sample supplied with the E2P inducing inhibitor orthovanadate and found about 11% of particles in an E2P conformation. This is rationalised as a residual fraction of unphosphorylated, non-inhibited, protein in the sample, but the sample is not actually tested for a residual unphosphorylated fraction or residual activity. Please provide some further experimental evidence to support this statement (?) and/or tone down the text with an appropriate caveat. Indeed, referee 2 suggests that perhaps the E1-P configuration could be a high-energy state rather than an off-cycle state, and/or there is also the possibility that this state is biased by the use of detergents, i.e., is there any evidence to support the off-cycle state can be formed in a lipid environment.

*Reviewer #1 (Recommendations for the authors):*

The current paper is well written and experiments and adequately described. The novel nucleotide-free E1 apo tight and E1-P tight states have not been seen in other P-type ATPases. How can the authors be sure that these tight states are not just a detergent artefact and less compacted states would be seen in a membrane environment, i.e comparative structure in nanodiscs?

*Reviewer #2 (Recommendations for the authors):*

– It has become somewhat custom in the P-type ATPase field to use the unhyphenated state descriptors ('E1P' and 'E2P') or to use a tilde ('E1~P') for the covalently phosphorylated states, whereas the hyphenated descriptors are for the transition states of phosphorylation and dephosphorylation, usually captured by planar mimics such as AlFx or VO3. Associated ligands are indicated by a dot, eg E1·ATP or E2·Pi. Here, the authors use hyphens when they mean covalently phosphorylated states, which could be confusing. I hence suggest to stick to the more common nomenclature, or at least ensure that a clear distinction is made between covalent phosphorylation and transition of phosphorylation/dephosphorylation.

– The terms 'state' and 'conformation' are used somewhat interchangeably in the manuscript, and this should be carefully revised. Different catalytic states can have the same conformation, eg. E1·ATP and E1P·ADP do not have any significant conformational differences, but are different catalytic states.

– Where ever structures are compared, relevant RMSD values should be given, to give a quantitative measure of structural deviation.

– The term 'ATP hydrolysis' is used in several instances, where actually 'phosphorylation from ATP' is meant. Phosphorylation needs to be distinguished from ATP hydrolysis, the latter being only complete with π release at the end of the cycle. In fact, the word "hydrolysis" implies an attack by water, and this attack occurs on E2P, and not in connection with phosphorylation of E1.

– P5 line 115: the claim that the 10 maps cover the entire conformational cycle is not true and should be revised. There is no E1P structure (without ADP), and also no E2 structure (post-dephosphorylation).

– P8 line 184 – 188: as stated in the public review, I find it likely that this is not ATP but product-inhibited ADP is bound – which would also explain why the structure is more open, due to a lack of N-P contacts mediated by the γ phosphate. It also seems more likely that such an E1.ADP state persists for long enough to be captured on the grid. While there is no easy proof for one or the other, this possibility should at least be discussed in the paper, especially given the weak map data for the γ phosphate.

– P10 line 232: the claim of some percentage residual active protein to explain the captured 11% E2P particles is not very strong without experimental evidence. Ideally, this should be tested on the actual sample, rather than referenced from another paper.

– P27 line 616: Why was an E1 conformation used as a reference for the 3D classification of the orthovanadate data? Would that not introduce a bias towards E1-like particles over E2-like ones? If so, this would be problematic because the low fraction of E2-like particles is used to draw a functional conclusion. How is the particle distribution, if an E2 structure is used as a reference instead?

– In addition to the point above, there is no information about how the orthovanadate solution was prepared. If orthovanadate is simply dissolved in water, there will be a mixture of mono- and decavanadate, leaving the true orthovanadate concentration unknown and overestimated.

*Reviewer #3 (Recommendations for the authors):*

Overall: The paper feels slightly repetitive, the text could probably be streamlined.

55: CBS, there is also a regulatory K site in many P-type ATPases, so maybe specify it is the ion binding site region in the TMD you refer to.

131, for example: Is RT, allowed by the journal or should the temperature be specified?

145: stating that it is E1 here is unnecessary, since you later say it is E1P-ADP. It can just be removed and the paragraph still works fine.

157: "an E1P-conformation".

202: E1late is subscript, but not E1P tight. Why?

225: Figure 2 —figure supplement 3H does not exist.

270, Figure 3C does not exist.

Figure 6, make the ions larger and the ion transport direction more obvious.

434: "lower negative charge".

494: A short summary of the previously described methods would be helpful, not sure if such statements are allowed by the journal.

505: How was orthovanadate prepared.

544: When writing "described above", does this refer to the pre-processing described below at 571?

716 and 733: should this be Table 2?

763: Table 4 is only referenced on the title page (Page 1). It should be referenced somewhere else as well.

1125: Figure 5 —figure supplement 1 A and D are never referenced.

---

## [Author Response]

Essential revisions:1. At the heart of this study is the comparison of the newly determined KdpFABC structures with previously published ones (of which there are already 10). Yet, an overview of the already determined structures is missing. There are also no RMSD calculations to illustrate the magnitude of any structural deviations, which makes it hard to judge the true level of novelty/deviation from known structures. Please revise and report the r.m.s.d. differences and superimposition and, where required, highlight structure features consistent with the novel states. This is of particular importance for the E1-P·ATP tight structure, which is reported as novel and yet it's unclear how similar this structure is to the previously published crystal structure (5MRW). There also seemed to be few changes in the TM segments that one would expect for inward and outward-facing states.

This is a very valid point and we thank the reviewers for bringing it up. To provide a better overview and appreciation of conformational similarities and significant differences we have calculated RMSDs between all available structures of KdpFABC. They are summarised in the new Table 1 – Table Supplement 2. We have included individual rmsd values, whenever applicable and relevant, in the respective sections in the text and figures. We note that the RMSDs were calculated only between the cytosolic domains (KdpB N,A,P domains) after superimposition of the full-length protein on KdpA, which is rigid across all conformations of KdpFABC (see description in material and methods lines 738-745 or the caption to Table 1 – Table Supplement 2). We opted to not indicate the RMSD calculated between the full-length proteins, as the largest part of the complex does not undergo large structural changes (see Figure 1 —figure supplement 1, the transmembrane region of KdpB as well as KdpA, KdpC and KdpF show relatively small to no rearrangements compared to the cytosolic domains), and would otherwise obscure the relevant RMSD differences discussed here.

Notably, the transmembrane region of KdpB in the X-ray structure is displaced relative to the rest of the complex when compared to the arrangement found in any of the other 18 cryo-EM structures, which all align well in the TMD. These deviations make the crystal structure somewhat of an outlier, and might be a consequence of the crystal packing. For completeness, we have included an additional RMSD that was calculated between structures when aligned on the TMD of KdpB rather than KdpA (see description in material and methods lines 738-745 or the caption to Table 1 – Table Supplement 2). From the resulting RMSD of ca. 5 Å to the cytosolic domains of the E1P tight structures determined here, we conclude that these structures do have a certain similarity but are not identical. In particular not in the relative orientation of the cytosolic domains to the rest of the complex. We hope that including the RMSD in the text and separately highlighting the important features of the E1P tight state in the section “E1P tight is the consequence of an impaired E1P/E2P transition” makes the story now more conclusive.

To highlight the (comparatively small) changes in the TMD, we have expanded Table 1 —figure supplement 1 to include panels showing the outward-open half-channel in the E1 states with a constriction at the KdpA/KdpB interface and the inward-open half-channel in the E2 states. The largest observable rearrangements do however take place in the cytosolic domains. This is an absolute agreement with previous studies, which focused more on the transition occurring within the transmembrane region during the transport cycle (Stock et al., Nature Communication 2018; Silberberg et al., Nature Communication 2021; Sweet et al., PNAS 2021).

2. The paper suggests that KdpFABC cannot undergo the transition from the E1P tight to E2P and gets stuck in this dead-end 'off cycle' state. To test this, the authors analysed an S162-P sample supplied with the E2P inducing inhibitor orthovanadate and found about 11% of particles in an E2P conformation. This is rationalised as a residual fraction of unphosphorylated, non-inhibited, protein in the sample, but the sample is not actually tested for a residual unphosphorylated fraction or residual activity. Please provide some further experimental evidence to support this statement (?) and/or tone down the text with an appropriate caveat. Indeed, referee 2 suggests that perhaps the E1-P configuration could be a high-energy state rather than an off-cycle state, and/or there is also the possibility that this state is biased by the use of detergents, i.e., is there any evidence to support the off-cycle state can be formed in a lipid environment.

In addition to the reference to Sweet et al. 2020, which reports the same residual activity, we have added a panel in Figure 2 – Supplementary Figure 5. Here, we show the residual ATPase activity of WT KdpFABC compared to KdpFAB_S162A_C, both purified from *E. coli* LB2003 cells with high potassium concentration, which is identical to the protein production and purification for the cryo-EM samples. The residual ATPase activity seen for the WT is ca. 14% of the uninhibited sample KdpFAB_S162A_C. This correlates with the E2-P fraction observed in the orthovanadate sample, and as reported before most likely can be attributed to residual non-phosphorylated Ser162.

We believe it is highly unlikely that the reported E1-P tight state represents an on-cycle high-energy E1P intermediate. For one, we observe a relaxation of electrostatic strains in this structure, in particular when compared to the obtained E1P ADP state. By contrast, the E1P should be the most energetically unfavourable state possible to ensure the rapid transition to the E2P state. As such, this state should be a transient state, making it less likely to be obtainable structurally as an accumulated state. Additionally, the association of the N domain with the A domain in the tight conformation, which would have to be reverted, would be a surprising intermediary step in the transition from E1P to E2P. Altogether, the here reported E1P tight state most likely represents an off-cycle state.

We cannot fully exclude the possibility that the detergent surrounding might have an effect on the structure of KdpFABC. However, we consider it unlikely that the tight states are detergent artefacts for the following reasons:

– As a simple test for the viability of the detergent-solubilised structures in a lipid bilayer, atomistic MD simulations (3x 50 ns for each of the 8 states) were performed to determine structural fluctuations from the input PDB. High deviation from the input would indicate structural elements that are either built in a manner that is energetically unfavourable, or that are observed as a consequence of the detergent micelle but are not compatible with the membrane bilayer. In our structures, we saw a low amount of structural fluctuations (low RMSDs); in fact, the tight states show among the lowest RMSDs relative to the input structures, indicating that they are likely among the most favourable conformations in a lipid membrane, so are not simply adopted due to detergent solubilization.

– All cryo-EM structures show clear non-protein densities around the transmembrane region, which are unlikely detergent and in some cases could be even confirmed to be cardiolipin. Thus, the detergent structures obtained for KdpFABC are not (entirely) delipidated.

– The domain rearrangement most relevant to the tight state is that of the N domain, which is farthest from the TM domain and is least likely to be affected by the detergent micelle.

– We were able to observe E1 apo open states, which are the least compacted conformations observed. Their presence contradicts the notion that the detergent favours/forces more compact folds.

– The two turnover samples without and with the inhibitory phosphorylation were prepared under identical conditions. If the tight state were detergent artefacts, we would expect it to be observable in both samples.

– The E1-P tight state we observe coincides with the point in the Post-Albers cycle at which KdpFABC is stalled, as was determined previously using biochemical methods (see Sweet et al., 2020, *eLife*)

– A magnitude of current cryo-EM structures obtained by our group and others (total 18) have all been determined in detergent. They all align well with the expected states of a P-type ATPase transport cycle, and have thus never raised a doubt of displaying any detergent artefact.

Altogether, we believe the reviewers will agree that the effort required to obtain a structure of KdpFABC in nanodiscs is too time consuming and would unlikely provide a different result, as elaborated above.

Reviewer #1 (Recommendations for the authors):The current paper is well written and experiments and adequately described. The novel nucleotide-free E1 apo tight and E1-P tight states have not been seen in other P-type ATPases. How can the authors be sure that these tight states are not just a detergent artefact and less compacted states would be seen in a membrane environment, i.e comparative structure in nanodiscs?

We cannot fully exclude the possibility that the detergent surrounding might have an effect on the structure of KdpFABC. However, we consider it unlikely that the tight states are detergent artefacts for the following reasons:

– As a simple test for the viability of the detergent-solubilised structures in a lipid bilayer, atomistic MD simulations (3x 50 ns for each of the 8 states) were performed to determine structural fluctuations from the input PDB. High deviation from the input would indicate structural elements that are either built in a manner that is energetically unfavourable, or that are observed as a consequence of the detergent micelle but are not compatible with the membrane bilayer. In our structures, we saw a low amount of structural fluctuations (low RMSDs); in fact, the tight states show among the lowest RMSDs relative to the input structures, indicating that they are likely among the most favourable conformations in a lipid membrane, so are not simply adopted due to detergent solubilization.

– All cryo-EM structures show clear non-protein densities around the transmembrane region, which are unlikely detergent and in some cases could be even confirmed to be cardiolipin. Thus, the detergent structures obtained for KdpFABC are not (entirely) delipidated.

– The domain rearrangement most relevant to the tight state is that of the N domain, which is farthest from the TM domain and is least likely to be affected by the detergent micelle.

– We were able to observe E1 apo open states, which are the least compacted conformations observed. Their presence contradicts the notion that the detergent favours/forces more compact folds.

– The two turnover samples without and with the inhibitory phosphorylation were prepared under identical conditions. If the tight state were detergent artefacts, we would expect it to be observable in both samples.

– The E1-P tight state we observe coincides with the point in the Post-Albers cycle at which KdpFABC is stalled, as was determined previously using biochemical methods (see Sweet et al., 2020, *eLife*)

– A magnitude of current cryo-EM structures obtained by our group and others (total 18) have all been determined in detergent. They all align well with the expected states of a P-type ATPase transport cycle, and have thus never raised a doubt of displaying any detergent artefact.

Altogether, we believe the reviewers will agree that the effort required to obtain a structure of KdpFABC in nanodiscs is too time consuming and would unlikely provide a different result, as elaborated above.

Reviewer #2 (Recommendations for the authors):– It has become somewhat custom in the P-type ATPase field to use the unhyphenated state descriptors ('E1P' and 'E2P') or to use a tilde ('E1~P') for the covalently phosphorylated states, whereas the hyphenated descriptors are for the transition states of phosphorylation and dephosphorylation, usually captured by planar mimics such as AlFx or VO3. Associated ligands are indicated by a dot, eg E1·ATP or E2·Pi. Here, the authors use hyphens when they mean covalently phosphorylated states, which could be confusing. I hence suggest to stick to the more common nomenclature, or at least ensure that a clear distinction is made between covalent phosphorylation and transition of phosphorylation/dephosphorylation.

The nomenclature has been adapted accordingly.

– The terms 'state' and 'conformation' are used somewhat interchangeably in the manuscript, and this should be carefully revised. Different catalytic states can have the same conformation, eg. E1·ATP and E1P·ADP do not have any significant conformational differences, but are different catalytic states.

This is a very valid comment, thanks for pointing it out. The nomenclature has been adapted so that every distinct position in the catalytic cycle of KdpFABC is referred to as a state rather than a conformation.

– Where ever structures are compared, relevant RMSD values should be given, to give a quantitative measure of structural deviation.

RMSD values have been added, with a focus on the RMSD between the cytosolic domains of different structures to more clearly show the structural differences, as the total RMSD would otherwise be skewed by the overall immobile transmembrane domain.

– The term 'ATP hydrolysis' is used in several instances, where actually 'phosphorylation from ATP' is meant. Phosphorylation needs to be distinguished from ATP hydrolysis, the latter being only complete with π release at the end of the cycle. In fact, the word "hydrolysis" implies an attack by water, and this attack occurs on E2P, and not in connection with phosphorylation of E1.

The term “hydrolysis” has been rephrased wherever it was used.

– P5 line 115: the claim that the 10 maps cover the entire conformational cycle is not true and should be revised. There is no E1P structure (without ADP), and also no E2 structure (post-dephosphorylation).

This is correct, and we have adjusted the phrasing to “close to the entire conformational cycle” or “the entire KdpFABC conformational cycle except the highly transient E1P state after ADP release and E2 state after dephosphorylation.”

– P8 line 184 – 188: as stated in the public review, I find it likely that this is not ATP but product-inhibited ADP is bound – which would also explain why the structure is more open, due to a lack of N-P contacts mediated by the γ phosphate. It also seems more likely that such an E1.ADP state persists for long enough to be captured on the grid. While there is no easy proof for one or the other, this possibility should at least be discussed in the paper, especially given the weak map data for the γ phosphate.

We have added this possible interpretation of the structure to the manuscript, but have maintained the previous assignment of E1·ATP_early_, which we still consider the most likely state for the following reasons:

1. While the density for the γ-phosphate is weaker than the α- and β-phosphates, we observed the same effect in our previously published E1·ATP states [7NNL] and [7NNP], which were prepared with AMPPCP and adopt the fully closed E1·ATP state.

2. We used 5 mM ATP in the sample preparation, which is a high excess over the protein. Therefore, ATP should displace any bound ADP, which we expect to be present in a much lower concentration as it is only formed through hydrolysis by the protein.

– P10 line 232: the claim of some percentage residual active protein to explain the captured 11% E2P particles is not very strong without experimental evidence. Ideally, this should be tested on the actual sample, rather than referenced from another paper.

In addition to the reference, we have added a panel in Figure 2—figure supplement Figure 5, in which we show the residual ATPase activity of WT KdpFABC compared to KdpFAB_S162A_C, both purified from *E. coli* LB2003 cells, which is identical to the protein production and purification for the cryo-EM samples. The residual ATPase activity is ca. 14% of the uninhibited sample, which correlates with the E2-P fraction in the orthovanadate sample.

– P27 line 616: Why was an E1 conformation used as a reference for the 3D classification of the orthovanadate data? Would that not introduce a bias towards E1-like particles over E2-like ones? If so, this would be problematic because the low fraction of E2-like particles is used to draw a functional conclusion. How is the particle distribution, if an E2 structure is used as a reference instead?

This is a very valid point, which we always pay attention to. We have revisited the image processing and can confirm that, when using a E2 reference, the number of particles assigned to an E2 state class barely changes (less than 1% difference) and that the E1P-tight state remains the dominant class. This statement is now also included in the figure legend of Figure 2 —figure supplement 4.

– In addition to the point above, there is no information about how the orthovanadate solution was prepared. If orthovanadate is simply dissolved in water, there will be a mixture of mono- and decavanadate, leaving the true orthovanadate concentration unknown and overestimated.

The preparation of orthovanadate was added to the methods section: “Orthovanadate was prepared by dissolving in water, adjusting the pH, and subsequent heating and pH adjustment iterations until no further discoloration was observable.”

Reviewer #3 (Recommendations for the authors):Overall: The paper feels slightly repetitive, the text could probably be streamlined.55: CBS, there is also a regulatory K site in many P-type ATPases, so maybe specify it is the ion binding site region in the TMD you refer to.

We have now denoted it as canonical substrate binding site (CBS) to clarify that this is the binding site for the transported ion.

131, for example: Is RT, allowed by the journal or should the temperature be specified?

We are not aware of a rule against using RT by *eLife*. To be certain, we have specified the temperature (for turnover conditions samples were incubated at 24 °C).

145: stating that it is E1 here is unnecessary, since you later say it is E1P-ADP. It can just be removed and the paragraph still works fine.

This has been implemented.

157: "an E1P-conformation".

This has been implemented.

202: E1late is subscript, but not E1P tight. Why?

For us, the early is a denomination for a substate within the E1·ATP state, since the conformation and structural features are overall very similar, while we define “E1P tight” as a completely separate state based on the distinct conformation.

225: Figure 2 —figure supplement 3H does not exist.

This has been corrected to Figure 2 —figure supplement 4H.

270, Figure 3C does not exist.

This has been corrected to Figure 3B.

Figure 6, make the ions larger and the ion transport direction more obvious.

Improved.

434: "lower negative charge".

This has been implemented.

494: A short summary of the previously described methods would be helpful, not sure if such statements are allowed by the journal.

A very brief description of the purification protocol has been added.

505: How was orthovanadate prepared.

The preparation of orthovanadate was added to the methods section: “Orthovanadate was prepared by dissolving in water, adjusting the pH, and subsequent heating and pH adjustment iterations until no further discoloration was observable.”

544: When writing "described above", does this refer to the pre-processing described below at 571?

Yes, and the pre-processing description has been shifted to the first sample described.

716 and 733: should this be Table 2?

Yes, this has been changed.

763: Table 4 is only referenced on the title page (Page 1). It should be referenced somewhere else as well.

A reference has been added at the beginning of the Methods section.

1125: Figure 5 —figure supplement 1 A and D are never referenced.

These panels have now been referenced.